# SAMPLING WITH MOLLIFIED INTERACTION ENERGY DESCENT

**Lingxiao Li**
MIT CSAIL
lingxiao@mit.edu

**Qiang Liu**
University of Texas at Austin
lqiang@cs.utexas.edu

**Anna Korba**
CREST, ENSAE, IP Paris
anna.korba@ensae.fr

**Mikhail Yurochkin**
IBM Research, MIT-IBM Watson AI Lab
mikhail.yurochkin@ibm.com

**Justin Solomon**
MIT CSAIL
jsolomon@mit.edu

## ABSTRACT

Sampling from a target measure whose density is only known up to a normalization constant is a fundamental problem in computational statistics and machine learning. In this paper, we present a new optimization-based method for sampling called *mollified interaction energy descent* (MIED). MIED minimizes a new class of energies on probability measures called *mollified interaction energies* (MIEs). These energies rely on mollifier functions—smooth approximations of the Dirac delta originated from PDE theory. We show that as the mollifier approaches the Dirac delta, the MIE converges to the chi-square divergence with respect to the target measure and the minimizers of MIE converge to the target measure. Optimizing this energy with proper discretization yields a practical first-order particle-based algorithm for sampling in both unconstrained and constrained domains. We show experimentally that for unconstrained sampling problems, our algorithm performs on par with existing particle-based algorithms like SVGD, while for constrained sampling problems our method readily incorporates constrained optimization techniques to handle more flexible constraints with strong performance compared to alternatives.

## 1 INTRODUCTION

Sampling from an unnormalized probability density is a ubiquitous task in statistics, mathematical physics, and machine learning. While Markov chain Monte Carlo (MCMC) methods (Brooks et al., 2011) provide a way to obtain unbiased samples at the price of potentially long mixing times, variational inference (VI) methods (Blei et al., 2017) approximate the target measure with simpler (e.g., parametric) distributions at a lower computational cost. In this work, we focus on a particular class of VI methods that approximate the target measure using a collection of interacting particles. A primary example is Stein variational gradient descent (SVGD) proposed by Liu & Wang (2016), which iteratively applies deterministic updates to a set of particles to decrease the KL divergence to the target distribution.

While MCMC and VI methods have found great success in sampling from unconstrained distributions, they often break down for distributions supported in a constrained domain. Constrained sampling is needed when the target density is undefined outside a given domain (e.g., the Dirichlet distribution), when the target density is not integrable in the entire Euclidean space (e.g., the uniform distribution), or when we only want samples that satisfy certain inequalities (e.g., fairness constraints in Bayesian inference (Liu et al., 2021)). A few recent approaches (Brubaker et al., 2012; Byrne & Girolami, 2013; Liu & Zhu, 2018; Shi et al., 2021) extend classical sampling methods like Hamiltonian Monte Carlo (HMC) or SVGD to constrained domains. These extensions, however, typically contain expensive numerical subroutines like solving nonlinear systems of equations and require explicit formulas for quantities such as Riemannian metric tensors or mirror maps to be derived on a case-by-case basis from the constraints.

We present an optimization-based method called *mollified interaction energy descent* (MIED) that minimizes *mollified interaction energies* (MIEs) for both unconstrained and constrained sampling. An MIE takes the form of a double integral of the quotient of a *mollifier*—smooth approximation of $\delta_0$, the Dirac delta at the origin—over the target density properly scaled. Intuitively, minimizing an MIE balances two types of forces: attractive forces that drive the current measure towards the target density, and repulsive forces from the mollifier that prevents collapsing. We show that as the mollifier converges to $\delta_0$, the MIE converges to the $\chi^2$ divergence to the target measure up to an additive constant (Theorem 3.3). Moreover, the MIE $\Gamma$-converges to $\chi^2$ divergence (Theorem 3.6), so that minimizers of MIEs converge to the target measure, providing a theoretical basis for sampling by minimizing MIE.

While mollifiers can be interpreted as kernels with diminishing bandwidths, our analysis is fundamentally different from that of SVGD where a fixed-bandwidth kernel is used to define a reproducing kernel Hilbert space (RKHS) on which the Stein discrepancy has a closed-form (Gorham & Mackey, 2017). Deriving a version of the Stein discrepancy for constrained domains is far from trivial and requires special treatment (Shi et al., 2021; Xu, 2021). In contrast, our energy has a unified form for constrained and unconstrained domains and approximates the $\chi^2$ divergence as long as the bandwidth is sufficiently small so that short-range interaction dominates the energy: this idea of using diminishing bandwidths in sampling is under-explored for methods like SVGD.

Algorithmically, we use first-order optimization to minimize MIEs discretized using particles. We introduce a log-sum-exp trick to neutralize the effect of arbitrary scaling of the mollifiers and the target density; this form also improves numerical stability significantly. Since we turn sampling into optimization, we can readily apply existing constrained sampling techniques such as reparameterization using a differentiable (not necessarily bijective) map or the dynamic barrier method by Gong & Liu (2021) to handle generic differentiable inequality constraints. Our method is effective as it only uses first-order derivatives of both the target density and the inequality constraint (or the reparameterization map), enabling large-scale applications in machine learning (see e.g. Figure 4). For unconstrained sampling problems, we show MIED achieves comparable performance to particle-based algorithms like SVGD, while for constrained sampling problems, MIED demonstrates strong performance compared to alternatives while being more flexible with constraint handling.

## 2 RELATED WORKS

**KL gradient flow and its discretization for unconstrained sampling.** The Wasserstein gradient flow of the Kullback-Leibler (KL) divergence has been extensively studied, and many popular sampling algorithms can be viewed as discretizations of the KL-divergence gradient flow. Two primary examples are Langevin Monte Carlo (LMC) and Stein variational gradient descent (SVGD). LMC simulates Langevin diffusion and can be viewed as a forward-flow splitting scheme for the KL-divergence gradient flow (Wibisono, 2018). At each iteration of LMC, particles are pulled along $-\nabla \log p$ where $p$ is the target density, while random Gaussian noise is injected, although a Metropolis adjusting step is typically needed for unbiased sampling. In contrast with LMC, SVGD is a deterministic algorithm that updates a collection of particles using a combination of an attractive force involving $-\nabla \log p$ and a repulsive force among the particles; it can be viewed as a kernelized gradient flow of the KL divergence (Liu, 2017) or of the $\chi^2$ divergence (Chewi et al., 2020). The connection to the continuous gradient flow in the Wasserstein space is fruitful for deriving sharp convergence guarantees for these sampling algorithms (Durmus et al., 2019; Balasubramanian et al., 2022; Korba et al., 2020; Salim et al., 2022).

**Sampling in constrained domains.** Sampling in constrained domains is more challenging compared to the unconstrained setting. Typical solutions are rejection sampling and reparameterization to an unconstrained domain. However, rejection sampling can have a high rejection rate when the constrained domain is small, while reparameterization maps need to be chosen on a case-by-case basis with a determinant-of-Jacobian term that can be costly to evaluate. Brubaker et al. (2012) propose a constrained version of HMC for sampling on implicit submanifolds, but their algorithm is expensive as they need to solve a nonlinear system of equations for every integration step in each step of HMC. Byrne & Girolami (2013) propose geodesic Hamiltonian Monte Carlo for sampling on embedded manifolds, but they require explicit geodesic formulae. Zhang et al. (2020); Ahn & Chewi (2021) propose discretizations of the mirror-Langevin diffusion for constrained sampling

provided that a mirror map is given to capture the constraint. Similarly, Shi et al. (2021) propose mirror SVGD that evolves particles using SVGD-like updates in the dual space defined by a mirror map. While these two methods have strong theoretical convergence guarantees, they are limited by the availability of mirror maps that capture the constraints. Liu et al. (2021) extend SVGD to incorporate a population constraint over the samples obtained; their setting is different from ours where the constraint is applied to every sample.

**Pairwise interaction energies on particles.** Pairwise interaction energies on particles take the form $E(\{x_i\}_{i=1}^N) = \sum_{i \neq j} W(x_i, x_j)$ for a kernel $W(x, y)$. The gradient flow of $E$ on $N$ particles can give rise to phenomena like flocking and swarming, and a long line of mathematical research studies mean-field convergence of such particle gradient flow to continuous solutions of certain PDEs as $N \to \infty$ (Carrillo et al., 2014; Serfaty, 2020). Of particular interest to sampling, a separate line of works summarized by Borodachov et al. (2019) demonstrates that for the hypersingular Riesz kernel $W(x, y) = \|x - y\|_2^{-s}$ with sufficiently big $s$, minimizing $E$ over a compact set yields uniform samples as $N \to \infty$. Moreover, $W$ can depend on $p$ to obtain non-uniform samples distributed according to $p$. As the hypersingular Riesz kernel is not integrable, their analysis is entirely based on geometric measure theory and avoids any variational techniques. In contrast, we take a different approach by using integrable kernels in MIEs through mollifiers. This enables us to apply variational analysis to establish interesting connections between MIEs and $\chi^2$ divergence. We compare our formulation with theirs further in Appendix B. Joseph et al. (2019) propose to minimize the maximum of pairwise interaction akin to the interaction in Borodachov et al. (2019) without gradient information using a greedy algorithm. Recently, Korba et al. (2021) propose sampling via descending on kernel Stein discrepancy which can also be viewed as a type of interaction energy, but like SVGD, it is limited to unconstrained sampling and can be slow as higher-order derivatives of $\log p$ are required. Craig et al. (2022) propose approximating the $\chi^2$ divergence with a functional different from ours that also involves a mollifier. The resulting algorithm needs to evaluate an integral containing the target density over the whole domain at each step which can be costly. Consequently, their method is only applied to special target densities for which the integral has a closed form. In comparison, our method can handle general target densities with cheap updates in each step.

## 3  MOLLIFIED INTERACTION ENERGY

**Notation.** Let $X \subset \mathbf{R}^n$ be the domain we work in. We use $\mathcal{L}_n$ to denote the Lebesgue measure on $\mathbf{R}^n$ and write $\int f(x) \, \mathrm{d}x$ to denote integration with respect to $\mathcal{L}_n$. We will assume all measures are Borel. We use $\mathcal{H}_d$ to denote the $d$-dimensional Hausdorff measure. We use $\omega_N$ to denote a set of points $\{x_1, \ldots, x_N\} \subset \mathbf{R}^n$. For $x \in \mathbf{R}^n$, let $\delta_x$ be the Dirac delta measure at $x$ and $\delta_{\omega_N}$ be the empirical measure $\frac{1}{N} \sum_{k=1}^N \delta_{x_k}$. We denote $L^p(X) := \{f : X \to \mathbf{R} \text{ Lebesgue measurable with } \int_X |f(x)|^p \, \mathrm{d}x < \infty\}$, and for $f \in L^p(X)$ we let $\|f\|_p := \left(\int_X |f(x)|^p \, \mathrm{d}x\right)^{1/p}$. We use $\|f\|_\infty$ to denote the essential supremum of $|f|$. For $f, g : \mathbf{R}^n \to \mathbf{R}$, we define convolution $(f * g)(x) := \int f(y)g(x - y) \, \mathrm{d}y$ provided this integral exists. We use $C^k(X)$ to denote continuously $k$-differentiable functions and $C^\infty(X)$ to indicate the smooth functions. The space of continuous $k$-differentiable functions with compact support on $X$ is $C_c^k(X)$. We denote by $\mathcal{P}(X)$ the set of probability measures, and $\mathcal{P}_2(X)$ the set of probability measures with bounded second moments. We denote by $\mathcal{M}_{\mathrm{sign}}(X)$ the set of finite signed measures. Given a Lebesgue measurable map $\boldsymbol{T} : X \to X$ and $\mu \in \mathcal{P}_2(X)$, $\boldsymbol{T}_\# \mu$ is the pushforward measure of $\mu$ by $\boldsymbol{T}$.

**Setup.** The domain $X \subset \mathbf{R}^n$ where we constrain samples is assumed to be closed and full-dimensional, i.e., $\mathcal{L}_n(X) > 0$; the unconstrained case corresponds to $X = \mathbf{R}^n$. The target density, always denoted as $p$, is assumed to satisfy $p \in C^1(X)$, $p(x) > 0$ for all $x \in X$, and $0 < \int_X p(x) \, \mathrm{d}x < \infty$. Let $\mu^*$ be the target probability measure $\mu^*(B) := \int_B p(x) \, \mathrm{d}x / \int_X p(x) \, \mathrm{d}x$ for Borel $B \subset X$. Our goal is to sample from $\mu^*$.

### 3.1  MOLLIFIERS

Our framework relies on the notion of *mollifiers* originally introduced in Friedrichs (1944).

**Definition 3.1** (family of mollifiers). *We say $\{\phi_\epsilon\}_{\epsilon > 0} \subset C^1(\mathbf{R}^n)$ is a family of mollifiers if it satisfies*

(a) *For any $\epsilon > 0$, $x \in \mathbf{R}^n$, $\phi_\epsilon(x) \geq 0$ and $\phi_\epsilon(x) = \phi_\epsilon(-x)$.*
(b) *For any $\epsilon > 0$, $\|\phi_\epsilon\|_1 = 1$ and $\sup_{x \in \mathbf{R}^n} \phi_\epsilon(x) < \infty$.*
(c) *For any $\delta > 0$, $p \in \{1, \infty\}$, $\lim_{\epsilon \to 0} \left\| \mathbf{1}_{\mathbf{R}^n \setminus B_\delta(0)} \phi_\epsilon \right\|_p = 0$.*

As $\epsilon \to 0$, the distribution with density $\phi_\epsilon$ converges to $\delta_0$, and if $f \in L^p(\mathbf{R}^n)$ and is continuous, then convolution $\phi_\epsilon * f$ converges to $f$ both pointwise (Proposition A.4) and in $L^p$ (Proposition A.7).

Our definition is different from the one used in PDE theory (Hörmander, 2015), where some $\phi \in C_c^\infty(\mathbf{R}^n)$ is fixed and the family of mollifiers is generated by $\phi_\epsilon(x) := \epsilon^{-n} \phi(x/\epsilon)$. While in PDE theory mollifiers are typically used to improve the regularity of non-smooth functions, in our framework they are used to construct interaction energies that approximate the $\chi^2$ divergence.

We will use the following mollifiers. In the following definitions, we include normalizing constants $Z_\epsilon$ that ensure $\|\phi_\epsilon\| = 1$. We do not write out $Z_\epsilon$ explicitly as they might not admit clean forms and they are only relevant in theory but not in our practical algorithm. For $s > n$, the *s-Riesz family of mollifiers* is defined as $\phi_\epsilon^s(x) := (\|x\|_2^2 + \epsilon^2)^{-s/2}/Z_\epsilon^s$. The *Gaussian family of mollifiers* is defined as $\phi_\epsilon^g(x) := \exp\left(-\|x\|_2^2/2\epsilon^2\right)/Z_\epsilon^g$. The *Laplace family of mollifiers* is defined as $\phi_\epsilon^l(x) := \exp\left(-\|x\|_2/\epsilon\right)/Z_\epsilon^l$. Since $\{\phi_\epsilon^g\}$ and $\{\phi_\epsilon^l\}$ correspond to the densities of centered Gaussian and Laplace random variables, they satisfy Definition 3.1. Proposition A.1 proves that $\{\phi_\epsilon^s\}$ also satisfies Definition 3.1.

## 3.2 Mollified Interaction Energies

**Definition 3.2** (mollified interaction energy)**.** *Given a family of mollifiers $\{\phi_\epsilon\}_{\epsilon > 0}$ satisfying Definition 3.1, for each $\epsilon > 0$, define a symmetric kernel $W_\epsilon : \mathbf{R}^n \times \mathbf{R}^n \to [0, \infty]$ by*

$$W_\epsilon(x, y) := \begin{cases} \phi_\epsilon(x - y)(p(x)p(y))^{-1/2} & \text{if } x, y \in X \\ \infty & \text{otherwise.} \end{cases} \tag{1}$$

*Define the* mollified interaction energy *(MIE), $\mathcal{E}_\epsilon : \mathcal{P}(\mathbf{R}^n) \to [0, \infty]$, to be*

$$\mathcal{E}_\epsilon(\mu) := \iint W_\epsilon(x, y) \, d\mu(x) \, d\mu(y). \tag{2}$$

Intuitively, minimizing $\mathcal{E}_\epsilon(\mu)$ balances two forces: a repulsive force from $\phi_\epsilon(x - y)$ makes sure $\mu$ has a good spread, and an attractive force from $(p(x)p(y))^{-1/2}$ helps $\mu$ concentrate on high-density regions. The exponent $-1/2$ is chosen to balance the two forces so that, as we will show in Theorem 3.3, $\mathcal{E}_\epsilon(\mu)$ approximates the $\chi^2$ divergence with respect to $\mu^*$ for small $\epsilon$.

## 3.3 Convergence to $\chi^2$-divergence

Recall that the $\chi^2$-divergence between probability measures $P, Q$ is defined by

$$\chi^2(Q \parallel P) := \begin{cases} \int \left(\frac{dQ}{dP} - 1\right)^2 dP & \text{if } Q \ll P \\ \infty & \text{otherwise,} \end{cases} \tag{3}$$

where $Q \ll P$ denotes absolute continuity of $Q$ with respect to $P$ with density $dQ/dP$.

**Theorem 3.3.** *Suppose $\mu \in \mathcal{P}(\mathbf{R}^n)$ satisfies $\chi^2(\mu \parallel \mu^*) < \infty$. Then, $\mathcal{E}_\epsilon(\mu) < \infty$ for any $\epsilon > 0$. Furthermore,*

$$\lim_{\epsilon \to 0} \mathcal{E}_\epsilon(\mu) = \chi^2(\mu \parallel \mu^*) + 1.$$

We provide a succinct proof of Theorem 3.3 in Appendix A.2 using the theory of mollifiers developed in Appendix A.1. In Remark A.10 we discuss an extension of Theorem 3.3 to cases when $X$ is not full-dimensional; in particular Theorem 3.3 still holds when $X$ is "flat", i.e., has Hausdorff dimension $d < n$ and is contained in a $d$-dimensional linear subspace.

### 3.4 Convexity and Γ-convergence

We next study properties of the minimizers of MIEs: Does $\min_{\mu \in \mathcal{P}(X)} \mathcal{E}_\epsilon(\mu)$ admit a unique minimum? If so, do minima of $\{\mathcal{E}_\epsilon\}_{\epsilon > 0}$ converge to $\mu^*$ as $\epsilon \to 0$? In order to answer affirmatively to these questions, we will need the associated kernel $k_\epsilon(x, y) := \phi_\epsilon(x - y)$ to satisfy the following property.

**Definition 3.4** (i.s.p.d. kernel). *A symmetric lower semicontinuous (l.s.c.) kernel $K$ on $X \times X$ is integrally strictly positive definite (i.s.p.d.) if for every finite signed Borel measure $\nu$ on $X$, the energy $\mathcal{E}_K(\nu) := \iint K(x, y) \, d(\nu \times \nu)(x, y)$ is well-defined (i.e., the integrals over the negative and positive parts of $\nu$ are not both infinite), and $\mathcal{E}_K(\nu) \geq 0$ where the equality holds only if $\nu = 0$ on all Borel sets of $X$.*

For the mollifiers we consider, the associated kernel $k_\epsilon(x, y) = \phi_\epsilon(x - y)$ is i.s.p.d. on any compact set (Pronzato & Zhigljavsky (2021, Example 1.2)).

**Proposition 3.5.** *Suppose $X$ is compact and $k_\epsilon(x, y) := \phi_\epsilon(x - y)$ is i.s.p.d. on $X$. Then,*

- *(a) The kernel $W_\epsilon(x, y)$ defined in (1) also i.s.p.d on $X$.*
- *(b) The functional $\mathcal{E}_\epsilon$ is strictly convex on $\mathcal{M}_{\text{sign}}(X)$ and attains a unique minimum on $\mathcal{P}(X)$.*

We next show the convergence of minima of $\{\mathcal{E}_\epsilon\}$ to $\mu^*$ as a consequence of Γ-convergence of the sequence $\{\mathcal{E}_\epsilon\}$.

**Theorem 3.6.** *Suppose $k_\epsilon(x, y) := \phi_\epsilon(x - y)$ is i.s.p.d. on compact sets for every $\epsilon > 0$. Then we have Γ-convergence (Definition A.13) $\mathcal{E}_\epsilon \xrightarrow{\Gamma} \chi^2(\cdot \parallel \mu^*) + 1$ as $\epsilon \to 0$. In particular, if $X$ is compact, if we denote $\mu_\epsilon^* := \arg\min_{\mu \in \mathcal{P}(X)} \mathcal{E}_\epsilon(\mu)$, then $\mu_\epsilon^* \to \mu^*$ weakly and $\lim_{\epsilon \to 0} \mathcal{E}_\epsilon(\mu_\epsilon^*) = 1$.*

We prove Theorem 3.6 in Appendix A.3 using Fourier transforms and Bochner's theorem. Theorem 3.6 provides basis for minimizing $\mathcal{E}_\epsilon$ for a small $\epsilon$, since its unique minimum will be a good approximation of $\mu^*$.

### 3.5 Differential calculus of $\mathcal{E}_\epsilon$ in $\mathcal{P}_2(\mathbf{R}^n)$

We next study the gradient flow of $\mathcal{E}_\epsilon$ in Wasserstein space $\mathcal{P}_2(\mathbf{R}^n)$ for $X = \mathbf{R}^n$. Understanding the gradient flow of a functional often provides insights into the convergence of algorithms that simulates gradient flow with time discretization (Ambrosio et al., 2005, Chapter 11) or spatial discretization (Chizat, 2022). Proofs of this section are given in Appendix A.4.

Let $\mathcal{F} : \mathcal{P}_2(\mathbf{R}^n) \to \mathbf{R}$ be a functional. The *Wasserstein gradient flow* of $\mathcal{F}$ (Ambrosio et al., 2005, Definition 11.1.1) is defined as the solution $\{\mu_t\}_{t \geq 0}$ of the PDE: $\frac{\partial \mu_t}{\partial t} = \nabla \cdot (\mu_t w_{\mathcal{F}, \mu_t})$ where $w_{\mathcal{F}, \mu} \in L^2(\mu; \mathbf{R}^n)$ is a *Frechét subdifferential* of $\mathcal{F}$ at $\mu$. Intuitively, gradient flows capture the evolution of the variable being optimized if we were to do gradient descent[1] on $\mathcal{F}$ when the step sizes go to 0. We next show that the gradient flow of $\mathcal{E}_\epsilon$ agrees with that of $\chi^2(\cdot \parallel \mu^*)$ in the sense that their subdifferentials coincide as $\epsilon \to 0$.

**Proposition 3.7.** *Assume $\mu \in \mathcal{P}_2(\mathbf{R}^n)$ has density $q \in C_c^1(\mathbf{R}^n)^2$. Then any strong Frechét subdifferential (Definition A.20) of $\mathcal{E}_\epsilon$ at $\mu$ takes the form*

$$w_{\epsilon, \mu}(x) = 2\nabla \left( p(x)^{-1/2} (\phi_\epsilon * q/\sqrt{p})(x) \right), \text{ for } \mu\text{-a.e. } x \in \mathbf{R}^n. \tag{4}$$

*Moreover, if for sufficiently small $\epsilon$, $\phi_\epsilon$ has compact support independent of $\epsilon$, then*

$$\lim_{\epsilon \to 0} w_{\epsilon, \mu}(x) = w_{\chi^2, \mu}(x), \text{ for } \mu\text{-a.e. } x \in \mathbf{R}^n, \tag{5}$$

*where $w_{\chi^2, \mu}$ is a strong Frechét subdifferential of $\chi^2(\cdot \parallel \mu^*)$.*

While simulating $\chi^2$ divergence gradient flow is often intractable (Trillos & Sanz-Alonso, 2020, Section 3.3), our MIE admits a straightforward practical algorithm (Section 4).

---

[1] The more precise notion is of minimizing movements (Ambrosio et al., 2005, Chapter 2).

[2] We assume that $\mu$ has compact support because $W_\epsilon$ can be unbounded and cause integrability issues due to the presence of $(p(x)p(y))^{-1/2}$ term.

We next show that $\mathcal{E}_\epsilon$ is *displacement convex* at $\mu^* \in \mathcal{P}_2(\mathbf{R}^n)$ as $\epsilon \to 0$, obtaining a result similar to Korba et al. (2021, Corollary 4). This hints that gradient flow initialized near $\mu^*$ will have fast convergence.

**Proposition 3.8.** *Suppose $p \in C_c^2(\mathbf{R}^n)$. Suppose that for sufficiently small $\epsilon$, $\phi_\epsilon$ has compact support independent of $\epsilon$. Assume $k_\epsilon(x,y) := \phi_\epsilon(x-y)$ is i.s.p.d. Let $\boldsymbol{\xi} \in C_c^\infty(\mathbf{R}^n; \mathbf{R}^n)$ and $\mu_t := (\boldsymbol{I} + t\boldsymbol{\xi})_{\#}\mu^*$. Then $\lim_{\epsilon \to 0} \frac{\mathrm{d}^2}{\mathrm{d}t^2}\big|_{t=0} \mathcal{E}_\epsilon(\mu_t) \geq 0$.*

## 4 A PRACTICAL SAMPLING ALGORITHM

We now present a first-order particle-based algorithm for constrained and unconstrained sampling by minimizing a discrete version of (2). Substituting an empirical distribution for $\mu$ in (2) gives the discrete mollified interaction energy, for $N$ particles $\omega_N = \{x_1, \ldots, x_N\}$,

$$E_\epsilon(\omega_N) := \frac{1}{N^2} \sum_{i=1}^N \sum_{j=1}^N \phi_\epsilon(x_i - x_j)(p(x_i)p(x_j))^{-1/2}. \qquad (6)$$

Denote $\omega_{N,\epsilon}^* \in \arg\min_{\omega_N \subset X} E_\epsilon(\omega_N)$ and $\mu_\epsilon^* = \arg\min_{\mu \in \mathcal{P}_2(X)} \mathcal{E}_\epsilon(\mu)$, If $X$ is compact, by Borodachov et al. (2019, Corollary 4.2.9), we have weak convergence $\delta_{\omega_{N,\epsilon}^*} \to \mu_\epsilon^*$ as $N \to \infty$. If in addition $k_\epsilon(x,y) := \phi_\epsilon(x-y)$ is i.s.p.d., then by Theorem 3.6, we have weak convergence $\mu_\epsilon^* \to \mu^*$. This shows that minimizing (6) with a large $N$ and a small $\epsilon$ will result in an empirical distribution of particles that approximates $\mu^*$. Our sampling method, mollified interaction energy descent (MIED), is simply running gradient descent on (6) over particles (Algorithm 1) with an update that resembles the one in SVGD (see the discussion in Appendix C.1). Below we address a few practical concerns.

**Optimization in the logarithmic domain.** In practical applications, we only have access to the unnormalized target density $p$. The normalizing constant of the mollifier $\phi_\epsilon$ can also be hard to compute (e.g., for the Riesz family). While these normalization constants do not affect the minima of (6), they can still affect gradient step sizes during optimization. Moreover, $\phi_\epsilon$ can be very large when $\epsilon$ is small, and in many Bayesian applications $p$ can be tiny and only $\log p$ is numerically significant. To address these issues, we optimize the logarithm of (6) using the log-sum-exp trick (Blanchard et al., 2021) to improve numerical stability and to get rid of the arbitrary scaling of the normalizing constants:

$$\log E_\epsilon(\omega_N) := \log \sum_{i=1}^N \sum_{j=1}^N \exp\left(\log \phi_\epsilon(x_i - x_j) - \frac{1}{2}(\log p(x_i) + \log p(x_j))\right) - 2\log N. \qquad (7)$$

**Special treatment of the diagonal terms.** Since $\phi_\epsilon$ goes to the Dirac delta as $\epsilon \to 0$, the discretization of (2) on a neighborhood of the diagonal $\{(x,y) : x = y\}$ needs to be handled with extra care. In (6) the diagonal appears as $\sum_{i=1}^N \phi_\epsilon(0)p(x_i)^{-1}$ which can dominate the summation when $\epsilon$ is small and then $\phi_\epsilon(0)$ becomes too large. For the mollifiers that we consider, we use a different diagonal term $\sum_{i=1}^N \phi_\epsilon(h_i/\kappa_n)p(x_i)^{-1}$ where $h_i := \min_{j \neq i} \|x_i - x_j\|$ and $\kappa_n \geq 1$ is a constant depending only on the dimension $n$. Since $0 \leq \phi_\epsilon(h_i/\kappa_n) \leq \phi_\epsilon(0)$, the energy obtained will be bounded between (6) and the version of (6) without the diagonal terms. Hence by the proof of Theorem 4.2.2 of Borodachov et al. (2019), the discrete minimizers of $E_\epsilon$ still converge to $\mu_\epsilon^*$ as $N \to \infty$ and $\epsilon \to 0$. Empirically we found the choice $\kappa_n = (1.3n)^{1/n}$ works well for the Riesz family of mollifiers and we use the Riesz family primarily for our experiments.

**Constraint handling.** If $X \neq \mathbf{R}^n$, we need to minimize (7) subject to constraints $\omega_N = \{x_1, \ldots, x_N\} \subset X$. Since the constraint is the same for each particle $x_i$, we want our algorithm to remain parallelizable across particles.

We consider two types of constraints: (a) there exists a differentiable map $f : \mathbf{R}^n \to X$, with $\mathcal{L}_n(X \setminus f(\mathbf{R}^n)) = 0$; (b) the set $X$ is given by $\{x \in \mathbf{R}^n : g(x) \leq 0\}$ for a differentiable $g : \mathbf{R}^n \to \mathbf{R}$. For (a), we reduce the problem to unconstrained optimization in $\mathbf{R}^n$ using objective $\log E_\epsilon(f(\omega_N))$ with $\omega_N = \{x_1, \ldots, x_N\} \subset \mathbf{R}^n$ and $f(\omega_N) := \{f(x_1), \ldots, f(x_N)\}$. For (b), we apply the dynamic barrier method by Gong & Liu (2021) to particles in parallel; see Appendix C.2 for details and an extension to handling multiple constraints.

## 5 EXPERIMENTS

We compare MIED with recent alternatives on unconstrained and constrained sampling problems. Unless mentioned otherwise, we choose the $s$-Riesz family of mollifiers $\{\phi_\epsilon^s\}$ with $s = n+10^{-4}$ and $\epsilon = 10^{-8}$: we found minimizing the MIE with such mollifiers results in well-separated particles so that we can take $\epsilon$ to be very small as our theory recommends. This is not the case for the Gaussian or the Laplace family as setting $\epsilon$ too small can cause numerical issues even when particles are well-separated. In Appendix D.5, we compare different choices of $s$ on a constrained mixture distribution.

Unless mentioned otherwise: for SVGD (Liu & Wang, 2016), we use the adaptive Gaussian kernel as in the original implementation (adaptive kernels can be prone to collapse samples—see Appendix D.2); for KSDD (Korba et al., 2021), we use a fixed Gaussian kernel with unit variance. All methods by default use a learning rate of 0.01 with Adam optimizer (Kingma & Ba, 2014). The source code can be found at https://github.com/lingxiaoli94/MIED.

### 5.1 UNCONSTRAINED SAMPLING

**Gaussians in varying dimensions.** We first compare MIED with SVGD and KSDD on Gaussians of varying dimensions and with different numbers of particles. In Figure 1, we see that as the number of dimensions increases, MIED results in best samples in terms of $W_2$ distance (Wasserstein-2 distance computed using linear programming) with respect to $10^4$ i.i.d. samples, while SVGD yields lower energy distance (Székely & Rizzo, 2013). We think this is because MIED results in more evenly spaced samples (Figure 5) so the $W_2$ distance is lower, but it is biased since $\epsilon > 0$ so for energy distance SVGD performs better. More details can be found in Appendix D.1.

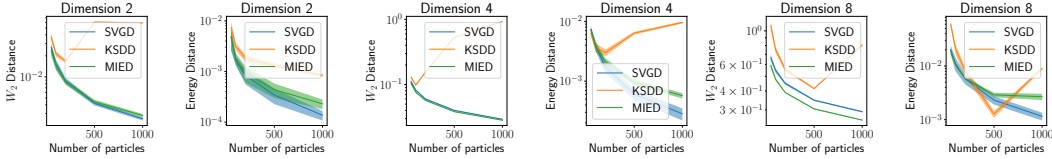

Figure 1: Gaussian experiments results in varying dimensions and different numbers of particles. For each method, we plot the metric ($W_2$ distance or energy distance) versus the number of particles, averaged over 10 trials (shaded region indicates standard deviation).

**Product of two Student's $t$-distributions.** Next, we consider a 2-dimensional distribution constructed as the product of two independent $t$-distributions with the degree of freedom $\nu = 2$ composed with a linear transform. Unlike Gaussians, Student's $t$-distributions have heavy tails. On the left of Figure 2, we visualize the results of each method with 1000 samples after 2000 iterations. MIED captures the heavy tail while SVGD fails. Quantitatively, on the right of Figure 2, while SVGD captures the distribution in $[-a, a]^2$ better for $a \le 3$, MIED yields lower metrics for a bigger $a$.

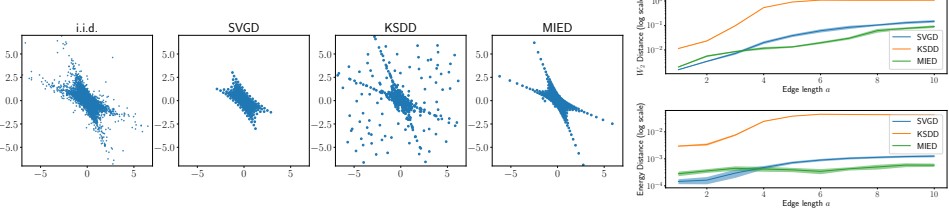

Figure 2: Left: visualization of samples from each method for the product of two Student's $t$-distribution (composed with a linear transform). Right: metrics of each method when restricting to $[-a, a]^2$. As $t$-distributions have heavy tails, if we draw i.i.d. samples from the true product distribution, a small number of them will have large norms, making the computation of metrics unstable. Thus we restrict both i.i.d. samples and the resulting samples from each method to $[-a, a]^2$ before computing the metrics for $a \in [2, 10]$.

**Bayesian logistic regression.** We compare MIED with SVGD and KSDD for the Bayesian logistic regression setting in Liu & Wang (2016). We further include a naïve baseline, independent particle descent (IPD), which simply runs gradient descent independently on each particle to maximize the posterior probability. In addition to using test accuracy as the metric, we include $W_2$ distance and energy distance between the samples from each method and $10^4$ samples from NUTS (Hoffman et al., 2014) after sufficient burn-in steps. We summarize the results in Table 5.1. All methods, including the naïve baseline IPD, are comparable in terms of test accuracy. In other words, accuracy is not a good metric for comparing the quality of posterior samples. Bayesian inference is typically preferred over maximum a posteriori estimation for its ability to capture uncertainty. When evaluating the quality of the uncertainty of the samples using distances between distributions, MIED provides the best approximation in terms of the $W_2$ distance and all methods (except IPD) are comparable in terms of the energy distance.

| Dataset | NUTS | IPD | SVGD | KSDD | MIED |
|---|---|---|---|---|---|
| banana ($d = 3$) | 0.55 | -6.14/-3.58/**0.55** | -7.81/-5.24/**0.55** | **-8.24/-5.76/0.55** | -7.37/-5.06/**0.55** |
| breast cancer ($d = 10$) | 0.64 | -1.51/-1.03/**0.60** | -1.62/-2.06/**0.60** | -1.71/**-2.23/0.60** | **-1.99/**-2.18/**0.60** |
| diabetis ($d = 9$) | 0.78 | -2.18/-1.55/**0.77** | -3.09/-3.42/**0.77** | -2.91/**-3.90/0.77** | **-3.11/**-3.13/**0.77** |
| flare solar ($d = 10$) | 0.59 | 3.30/2.65/0.48 | 6.91/4.09/0.52 | **1.77/-0.08/0.55** | 7.09/4.25/0.48 |
| german ($d = 21$) | 0.65 | -1.80/-1.25/**0.65** | -1.89/-2.63/0.64 | -1.27/**-2.83/0.65** | **-1.96/**-2.80/**0.65** |
| heart ($d = 14$) | 0.87 | -0.40/-0.56/**0.87** | -0.41/-1.50/**0.87** | -0.10/**-1.76/0.87** | **-0.92/**-1.67/**0.87** |
| image ($d = 19$) | 0.82 | 6.53/4.31/**0.83** | 7.17/4.01/**0.83** | 2.16/-0.50/0.82 | **1.14/-1.88/**0.82 |
| ringnorm ($d = 21$) | 0.77 | -3.82/-2.45/**0.77** | **-4.11/-5.98/0.77** | 1.07/-2.21/0.76 | -4.03/-5.70/**0.77** |
| splice ($d = 61$) | 0.85 | **-1.47/**-1.18/**0.85** | -1.22/**-2.65/0.85** | 2.04/-0.05/0.84 | 1.45/0.70/0.82 |
| thyroid ($d = 6$) | 0.84 | 1.95/0.53/**0.84** | 1.17/-0.00/**0.84** | 2.42/1.57/0.74 | **0.84/-0.37/0.84** |
| titanic ($d = 4$) | 0.40 | **-1.59/**-0.16/**0.40** | -0.46/-0.31/**0.40** | -0.63/-0.39/**0.40** | -1.00/**-0.45/0.40** |
| twonorm ($d = 21$) | 0.97 | -1.21/-1.13/**0.97** | -1.32/-2.78/**0.97** | 1.55/-0.62/**0.97** | **-1.44/-3.21/0.97** |
| waveform ($d = 22$) | 0.77 | -2.67/-1.87/**0.78** | -2.98/**-5.23/0.78** | -2.60/-4.18/0.77 | **-3.09/**-3.17/**0.78** |

Table 1: Bayesian logistic regression results with 1000 particles. We include the test accuracy for NUTS in the second column. Three numbers A/B/C in the following columns are logarithmic $W_2$ distance, logarithmic energy distance, and test accuracy. Bold indicates the best numbers. We use $80\%/20\%$ training/test split. All methods are run with identical initialization and learning rate 0.01. Results are reported after $10^4$ iterations.

## 5.2 Constrained sampling

**Uniform sampling in 2D.** We consider uniform sampling in the square $[-1, 1]^2$ with 500 particles. We reparameterize our particles using $\tanh$ to eliminate the constraint and show results with various choices of mollifiers—we always choose the smallest $\epsilon$ while the optimization remains stable. We compare our method with mirror LMC (Zhang et al., 2020) and SVMD/MSVGD by Shi et al. (2021) using entropic mirror map $\phi(\theta) = \sum_{i=1}^{n} \left( (1 + \theta_i) \log(1 + \theta_i) + (1 - \theta_i) \log(1 - \theta_i) \right)$. To demonstrate that SVGD and KSDD break down in constrained domains, we implement these two methods adapted to the constrained setting using the same reparameterization as our method. The initial particles are drawn uniformly from $[-0.5, 0.5]^2$. The left plot of Figure 3 shows that quantita-

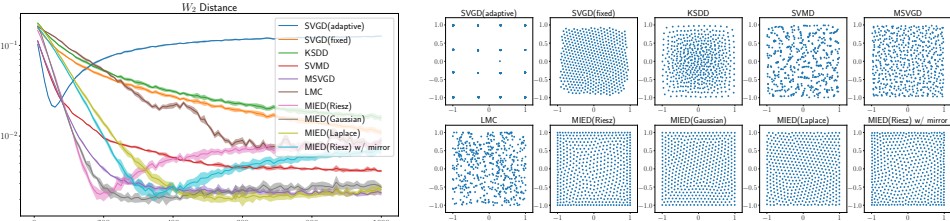

Figure 3: Convergence of metrics and visualization of samples for uniform sampling from a 2D box.

tively our method achieves much lower energy distance and $W_2$ distance (compared to a 5000 i.i.d. samples uniformly drawn in $[-1, 1]^2$.). On the right of Figure 3, we visualize the samples from each method. We see that samples from SVGD with an adaptive kernel collapse—we investigate this issue in Appendix D.2. Samples from SVGD with a fixed kernel size and KSDD are non-uniform (we choose kernel sizes that produce the best result). While SVMD creates locally clustered artifacts, MSVGD produces good results. For mirror LMC, the resulting samples are not evenly spaced,

resulting in worse $W_2$ distances. When using $\tanh$ as the reparameterization map, MIED produces decent results for all three families of mollifiers (Riesz, Gaussian, Laplace). When using the same entropic mirror map, MIED also produces good results. This highlights the flexibility of our method with constraint handling, whereas, for LMC and SVMD/MSVGD, the mirror map has to be chosen carefully to be the gradient of a strongly convex function while capturing the constraints.

In Appendix D.3, we further test SVMD/MSVGD with a different choice of the mirror map where they break down. In Appendix D.4, we conduct a similar comparison for sampling from a 20-dimensional Dirichlet distribution using the same setup as Shi et al. (2021). In scenarios where a good choice of a mirror map is available, SVMD/MSVGD can obtain better performance compared to MIED. In Appendix D.6, we conduct additional qualitative experiments for MIED, demonstrating its effectiveness for challenging constraints and multi-modal distributions.

**Fairness Bayesian neural networks.** We train fair Bayesian neural networks to predict whether the annual income of a person is at least $\$50,000$ with gender as the protected attribute using the *Adult Income* dataset (Kohavi et al., 1996). We follow the same setup as in Liu et al. (2021) where the dataset $\mathcal{D} = \{x^{(i)}, y^{(i)}, z^{(i)}\}_{i=1}^{|\mathcal{D}|}$ consists of feature vectors $x^{(i)}$, labels $y^{(i)}$ (whether the income is $\geq \$50,000$), and genders $z^{(i)}$ (protected attribute). The target density is taken to be the posterior of logistic regression with a two-layer Bayesian neural network $\hat{y}(\cdot; \theta)$ with weights $\theta$, and we put a standard Gaussian prior on each entry of $\theta$ independently. Given $t > 0$, the fairness constraint is $g(\theta) = (\text{cov}_{(x,y,z)\sim\mathcal{D}}[z, \hat{y}(x; \theta)])^2 - t \leq 0$. On the left of Figure 4, we plot the trade-off curve of the result obtained using our method and the methods from Liu et al. (2021) for $t \in \{10^{-5}, 10^{-4}, 0.0001, 0.001, 0.002, 0.005, 0.01\}$. Details can be found in Appendix D.7. Our method recovers a much larger Pareto front compared to the alternatives. On the right of Figure 4, we visualize the curves of the energy and the covariance versus the number of training iterations: as expected we see a smaller $t$ results in bigger MIE (lower log-likelihood) and smaller covariance between the prediction and the protected attribute.

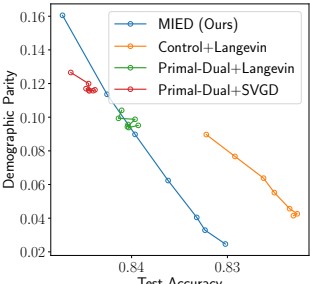 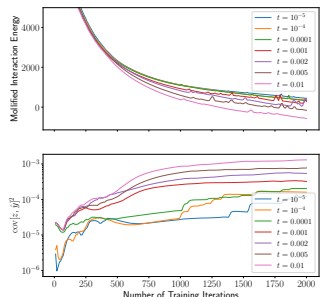

Figure 4: Left: trade-off curves of demographic parity versus accuracy on the test data for MIED and methods from Liu et al. (2021). Right: MIEs and $(\text{cov}_{(x,y,z)\sim\mathcal{D}}[z, \hat{y}(x; \theta)])^2$ (measured on the training data) versus the number of training iterations, for various $t$.

## 6 CONCLUSION

We present a new sampling method by minimizing MIEs discretized as particles for unconstrained and constrained sampling. This is motivated by the insight that MIEs converge to $\chi^2$ divergence with respect to the target measure as the mollifier parameter goes to 0. The proposed method achieves promising results on the sampling problems we consider.

Below we highlight three limitations. First, as discussed in Remark A.10, our theory only applies when the domain is full-dimensional or flat. Extending our theory to handle cases where the domain is an arbitrary $d$-rectifiable set is an important next step as it allows the handling of more complicated constraints such as nonlinear equality constraints. Secondly, when $\epsilon > 0$, the minimizer of MIE can be different from the target measure. Finding a way to debias MIE (e.g., like how Sinkhorn distances are debiased (Feydy et al., 2019)) is an interesting direction. Lastly, the connection between the minimizers of the discretized MIE (6) and those of the continuous MIE (2) is only established in the limit as $N \to \infty$. We hope to investigate how well the empirical distribution of particles minimizing (6) approximates the target measure when $N$ is finite as in Xu et al. (2022).

**Reproducibility statement.** We provide self-contained proofs in Appendix A for the theoretical results stated in the main text. The source code for all experiments can be found at https://github.com/lingxiaoli94/MIED.

**Acknowledgements** Qiang Liu would like to acknowledge the support of NSF CAREER 1846421 and ONR. The MIT Geometric Data Processing group acknowledges the generous support of Army Research Office grants W911NF2010168 and W911NF2110293, of Air Force Office of Scientific Research award FA9550-19-1-031, of National Science Foundation grants IIS-1838071 and CHS-1955697, from the CSAIL Systems that Learn program, from the MIT–IBM Watson AI Laboratory, from the Toyota–CSAIL Joint Research Center, from a gift from Adobe Systems, and from a Google Research Scholar award.

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

## A  DETAILED ANALYSIS

### A.1  PRELIMINARIES ON MOLLIFIERS

**Proposition A.1.** *For $s > n$, the $s$-Riesz family of mollifiers, defined as $\phi_\epsilon^s(x) := (\|x\|_2^2 + \epsilon^2)^{-s/2}/Z_\epsilon^s$, satisfies Definition 3.1.*

In order to prove Proposition A.1, we first prove a lemma.

**Lemma A.2.** *Let $b \geq 0$. Then for any $\epsilon > 0$,*

$$Z_\epsilon^s \int_{B_\epsilon(0)} \phi_\epsilon^s(y) \|y\|_2^b \, \mathrm{d}y = \mathcal{H}_{n-1}(S^{n-1}) \left( \int_0^1 \frac{t^{n+b-1}}{(t^2+1)^{s/2}} \right) \epsilon^{n+b-s}, \tag{8}$$

*where $\mathcal{H}_{n-1}(S^{n-1})$ is the volume of the $(n-1)$-dimensional sphere.*

*Furthermore, assuming $s > b + n$, then for any $\epsilon > 0, \delta > 0$,*

$$Z_\epsilon^s \int_{\mathbf{R}^n \setminus B_\delta(0)} \phi_\epsilon^s(y) \|y\|_2^b \, \mathrm{d}y \leq \mathcal{H}_{n-1}(S^{n-1}) \frac{\delta^{n+b-s}}{s-(n+b)}. \tag{9}$$

*Proof.* For (8), we compute

$$Z_\epsilon^s \int_{B_\epsilon(0)} \phi_\epsilon^s(y) \|y\|_2^b \, \mathrm{d}y = \int_{B_\epsilon(0)} \frac{\|y\|_2^b}{(\|y\|_2^2 + \epsilon^2)^{s/2}} = \mathcal{H}_{n-1}(S^{n-1}) \int_0^\epsilon \frac{r^{n+b-1}}{(r^2+\epsilon^2)^{s/2}} \, \mathrm{d}r$$

$$= \mathcal{H}_{n-1}(S^{n-1}) \int_0^1 \frac{(\epsilon t)^{n+b-1}}{(\epsilon^2 t^2 + \epsilon^2)^{s/2}} \epsilon \, \mathrm{d}t$$

$$= \mathcal{H}_{n-1}(S^{n-1}) \left( \int_0^1 \frac{t^{n+b-1}}{(t^2+1)^{s/2}} \, \mathrm{d}t \right) \epsilon^{n+b-s},$$

where we use substitution $r = \epsilon t$ on the second line.

If $s > b + n$, then

$$Z_\epsilon^s \int_{\mathbf{R}^n \setminus B_\delta(0)} \phi_\epsilon^s(0, y) \|y\|_2^b \, \mathrm{d}y = \int_{\mathbf{R}^n \setminus B_\delta(0)} \frac{\|y\|_2^b}{(\|y\|_2^2 + \epsilon^2)^{s/2}}$$

$$\leq \int_{\mathbf{R}^n \setminus B_\delta(0)} \frac{\|y\|_2^b}{\|y\|_2^s} = \mathcal{H}_{n-1}(S^{n-1}) \int_\delta^\infty r^{n-1+b-s}$$

$$= \mathcal{H}_{n-1}(S^{n-1}) \frac{\delta^{n+b-s}}{s-(n+b)},$$

where the last integral uses $s > b + n$. $\qquad\square$

*Proof of Proposition A.1.* It is clear that (a) of Proposition A.1 holds for $\phi_\epsilon^s(x)$. Since $\phi_\epsilon^s(x)$ is bounded we have $\phi_\epsilon^s \in L^\infty(\mathbf{R}^n)$. For any $\delta > 0$, we have, for $\epsilon \leq \delta$,

$$Z_\epsilon^s \int_{B_\delta(0)} \phi_\epsilon^s(x)\,\mathrm{d}x \geq Z_\epsilon^s \int_{B_\epsilon(0)} \phi_\epsilon^s(x)\,\mathrm{d}x = C\epsilon^{n-s},$$

where we use (8) with $b = 0$ and $C$ is a constant depending only on $n, s$. On the other hand, using (9) with $b = 0$,

$$Z_\epsilon^s \int_{\mathbf{R}^n \setminus B_\delta(0)} \phi_\epsilon^s(x)\,\mathrm{d}x \leq C'\delta^{n-s},$$

where $C'$ depends only on $n, s$. With $\delta = \epsilon$, we see that $Z_\epsilon^s \phi_\epsilon^s \in L^1(\mathbf{R}^n)$ so (b) is satisfied.

Since $\delta$ is fixed and $s > n$, we see that

$$\lim_{\epsilon \to 0} \frac{\int_{B_\delta(0)} \phi_\epsilon^s(x)\,\mathrm{d}x}{\int_{\mathbf{R}^n \setminus B_\delta(0)} \phi_\epsilon^s(x)\,\mathrm{d}x} \geq \lim_{\epsilon \to 0} \frac{C\epsilon^{n-s}}{C'\delta^{n-s}} = \infty.$$

Since $\int_{B_\delta(0)} \phi_\epsilon^s(x)\,\mathrm{d}x + \int_{\mathbf{R}^n \setminus B_\delta(0)} \phi_\epsilon^s(x)\,\mathrm{d}x = 1$, we have shown (c). $\qquad\square$

In the rest of this section, we assume $\{\phi_\epsilon\}_{\epsilon>0}$ is a family of mollifiers satisfying Definition 3.1.

**Lemma A.3.** *For any $p \in [1, \infty]$, for any $\delta > 0$, $\phi_\epsilon \in L^p(\mathbf{R}^n)$ and $\lim_{\epsilon \to 0} \left\|\mathbf{1}_{\mathbf{R}^n \setminus B_\delta(0)} \phi_\epsilon\right\|_p = 0.$ holds.*

*Proof.* Assume $p \notin \{1, \infty\}$ since both cases are covered in the assumptions. Then for any $\delta > 0$, by Hölder's inequality,

$$\|\phi_\epsilon\|_p^p = \left\|\phi_\epsilon \cdot \phi_\epsilon^{p-1}\right\|_1 \leq \|\phi_\epsilon\|_1 \left\|\phi_\epsilon^{p-1}\right\|_\infty = \|\phi_\epsilon\|_1 \|\phi_\epsilon\|_\infty^{p-1} < \infty.$$

Similarly,

$$\left\|\mathbf{1}_{\mathbf{R}^n \setminus B_\delta(0)} \phi_\epsilon\right\|_p^p = \left\|\mathbf{1}_{\mathbf{R}^n \setminus B_\delta(0)} \phi_\epsilon^p\right\|_1 = \left\|\phi_\epsilon \cdot \mathbf{1}_{\mathbf{R}^n \setminus B_\delta(0)} \phi_\epsilon^{p-1}\right\|_1$$

$$\leq \|\phi_\epsilon\|_1 \left\|\mathbf{1}_{\mathbf{R}^n \setminus B_\delta(0)} \phi_\epsilon^{p-1}\right\|_\infty = \left\|\mathbf{1}_{\mathbf{R}^n \setminus B_\delta(0)} \phi_\epsilon\right\|_\infty^{p-1}.$$

Letting $\epsilon \to 0$ and applying (c) gives $\lim_{\epsilon \to 0} \left\|\mathbf{1}_{\mathbf{R}^n \setminus B_\delta(0)} \phi_\epsilon\right\|_p = 0.$ $\qquad\square$

**Proposition A.4.** *Let $f \in L^p(\mathbf{R}^n)$, $p \in [1, \infty]$. Then for every $\epsilon > 0$, the integral*

$$\int f(x - y)\phi_\epsilon(y)\,\mathrm{d}y$$

*exists, so that $(f * \phi_\epsilon)(x)$ is finite. Moreover, if $f$ is continous at $x$, then*

$$\lim_{\epsilon \to 0} (f * \phi_\epsilon)(x) = f(x). \tag{10}$$

*Proof.* Observe that by Hölder's inequality, for $q$ such that $1/p + 1/q = 1$ (allowing infinity),

$$\int |f(x - y)\phi_\epsilon(y)|\,\mathrm{d}y \leq \|f\|_p \|\phi_\epsilon\|_q < \infty.$$

Hence $(f * \phi_\epsilon)(x)$ is integrable and finite.

For any $\epsilon > 0$, note that

$$|(f * \phi_\epsilon)(x) - f(x)| = \left|\int f(x - y)\phi_\epsilon(y)\,\mathrm{d}y - f(x)\right|.$$

Since $\int \phi_\epsilon(x)\,\mathrm{d}x = 1$, we have

$$|(f * \phi_\epsilon)(x) - f(x)| = \left| \int (f(x-y) - f(x))\,\phi_\epsilon(y)\,\mathrm{d}y \right| \leq \int |f(x-y) - f(x)| \phi_\epsilon(y)\,\mathrm{d}y.$$

Fix $t > 0$. Continuity of $f$ at $x$ implies there exists $\delta > 0$ such that $|f(x-y) - f(x)| < t$ for all $y \in B_\delta(0)$. Then

$$\int_{B_\delta(0)} |f(x-y) - f(x)| \phi_\epsilon(y)\,\mathrm{d}y \leq t \int_{B_\delta(0)} \phi_\epsilon(y)\,\mathrm{d}y \leq t.$$

On the other hand, by Hölder's inequality,

$$\int_{\mathbf{R}^n \setminus B_\delta(x)} |f(x-y) - f(x)| \phi_\epsilon(y)\,\mathrm{d}y \leq \|\tau_x f - f\|_p \|\mathbf{1}_{\mathbf{R}^n \setminus B_\delta(0)} \phi_\epsilon\|_q \leq 2\|f\|_p \|\mathbf{1}_{\mathbf{R}^n \setminus B_\delta(0)} \phi_\epsilon\|_q.$$

Hence

$$|(f * \phi_\epsilon)(x) - f(x)| \leq t + 2\|f\|_p \|\mathbf{1}_{\mathbf{R}^n \setminus B_\delta(0)} \phi_\epsilon\|_q.$$

By Lemma A.3, since $\|f\|_p < \infty$, taking $\epsilon \to 0$ we get, for any $t > 0$,

$$\limsup_{\epsilon \to 0} |(f * \phi_\epsilon)(x) - f(x)| \leq t.$$

Now let $t \to 0$ we get (10). $\qquad \square$

**Corollary A.5.** *Let $f \in L^p(\mathbf{R}^n)$, $p \in [1, \infty]$, be a continuous function. If $\{f_\epsilon\}$ is a sequence of functions that converge uniformly to $f$, then for every $x$,*

$$\lim_{\epsilon \to 0} (f_\epsilon * \phi_\epsilon)(x) = f(x). \tag{11}$$

*Proof.* Note that

$$|(\phi_\epsilon * f_\epsilon)(x) - f(x)| \leq |(\phi_\epsilon * f_\epsilon)(x) - (\phi_\epsilon * f)(x)| + |(\phi_\epsilon * f)(x) - f(x)|.$$

Proposition A.4 shows the second term goes to 0 as $\epsilon \to 0$. For the first term, we have

$$|(\phi_\epsilon * f_\epsilon)(x) - (\phi_\epsilon * f)(x)| = \left| \int (f_\epsilon(x-y) - f(x-y))\,\phi_\epsilon(y)\,\mathrm{d}y \right|$$
$$\leq \sup_x |f_\epsilon(x) - f(x)| \to 0$$

by uniform convergence. $\qquad \square$

**Lemma A.6.** *For $f \in L^p(\mathbf{R}^n)$, $p \in [1, \infty)$, we have*

$$\lim_{y \to 0} \|\tau_y f - f\|_p = 0,$$

*where we use $\tau_a f$ to denote the translated function $\tau_a f(x) := f(x-a)$.*

*Proof.* Fix $\epsilon > 0$. It is a standard fact that $C_c(\mathbf{R}^n)$ is dense in $L^p(\mathbf{R}^n)$. Hence there exists $g \in C_c(\mathbf{R}^n)$ such that $\|f - g\|_p < \epsilon$. Since $g$ is continuous with compact support, it is uniform continuous. Then there exists $\delta > 0$ with $|g(x-y) - g(x)| < \epsilon^{1/p}/\mathcal{L}_n(K)$ if $y \in B_\delta(x)$. Hence for such $y$ we have $\|\tau_y g - g\|_p^p < \epsilon$ with $\mathcal{L}_n(K) < \infty$. Thus

$$\|\tau_y f - f\|_p \leq \|\tau_y f - \tau_y g\|_p + \|\tau_y g - g\|_p + \|g - f\|_p \leq 3\epsilon.$$

$\qquad \square$

**Proposition A.7.** *Let $f \in L^p(\mathbf{R}^n)$, $p \in (1, \infty)$. Then for every $\epsilon > 0$,*

$$\|f * \phi_\epsilon\|_p \leq 3\|f\|_p.$$

*In particular, $f * \phi_\epsilon \in L^p(\mathbf{R}^n)$. Moreover,*

$$\lim_{\epsilon \to 0} \|f * \phi_\epsilon - f\|_p = 0.$$

*Proof.* For every $\epsilon > 0$, we have

$$\|f * \phi_\epsilon - f\|_p^p = \int \left| \int (f(x-y) - f(x))\phi_\epsilon(y)\,dy \right|^p dx$$

$$\leq \int \int |f(x-y) - f(x)|^p \phi_\epsilon(y)\,dy\,dx$$

$$= \int \left( \int |f(x-y) - f(x)|^p\,dx \right) \phi_\epsilon(y)\,dy$$

$$= \int \|\tau_y f - f\|_p^p \phi_\epsilon(y)\,dy,$$

where we use Jensen's inequality with the observation that $\phi_\epsilon(y)\,dy$ is a probability measure and Tonelli's theorem to exchange the order of integration. To show the first claim, note that,

$$\|f * \phi_\epsilon - f\|_p^p \leq \int \|\tau_y f - f\|_p^p \phi_\epsilon(y)\,dy \leq \int (2\|f\|_p)^p \phi_\epsilon(y)\,dy = (2\|f\|_p)^p.$$

Hence $\|f * \phi_\epsilon\|_p \leq \|f\|_p + \|f * \phi_\epsilon - f\|_p \leq 3\|f\|_p$.

For the second claim, by Lemma A.6, the function $y \mapsto \|\tau_y f - f\|_p^p$ is continuous at $y = 0$ with limit 0. Hence by Proposition A.4, we are done by taking $\epsilon \to 0$. $\square$

## A.2 CONVERGENCE TO $\chi^2$-DIVERGENCE

We start by giving an alternative formula for $\chi^2$ divergence with respect to the target measure.

**Lemma A.8.** *Define a functional* $\mathcal{E} : \mathcal{P}(\mathbf{R}^n) \to [0, \infty]$ *by*

$$\mathcal{E}(\mu) := \begin{cases} \int_X \frac{q(x)^2}{p(x)}\,dx & \text{if } d\mu(x) = q(x)\,d\mathcal{L}_n(x) \text{ and } \mu \ll \mu^* \\ \infty & \text{otherwise.} \end{cases} \tag{12}$$

*Then for every* $\mu \in \mathcal{P}(\mathbf{R}^n)$,

$$\mathcal{E}(\mu) = \chi^2(\mu \parallel \mu^*) + 1.$$

*Proof.* If $\mu$ is not absolutely continuous with respect to $\mu^*$, then both sides are infinity. Otherwise, $\mu(X) = 1$ and $\mu$ has some density $q$. We then compute

$$\chi^2(\mu \parallel \mu^*) + 1 = \int \left( \frac{d\mu}{d\mu^*}(x) - 1 \right)^2 d\mu^*(x) + 1 = \int_X \left( \frac{q(x)}{p(x)} - 1 \right)^2 p(x)\,dx + 1$$

$$= \int_X \frac{q(x)^2}{p(x)}\,dx - 2\int_X q(x)\,dx + 2 = \int_X \frac{q(x)^2}{p(x)}\,dx = \mathcal{E}(\mu).$$

$\square$

*Proof of Theorem 3.3.* Since $\chi^2(\mu \parallel \mu^*) < \infty$, we know $\mu \ll \mu^*$, so $\mu(X) = 1$. We let $q(x)$ be the density of $\mu$ which satisfies $q(x) = 0$ for $x \in \mathbf{R}^n \setminus X$. We compute using Tonelli's theorem (since our integrand is positive):

$$\mathcal{E}_\epsilon(\mu) = \iint \phi_\epsilon(x-y)(p(x)p(y))^{-1/2} q(x)q(y)\,dx\,dy$$

$$= \int \left( \phi_\epsilon * \frac{q}{\sqrt{p}} \right)(x) \frac{q}{\sqrt{p}}(x)\,dx = \int (\phi_\epsilon * g)(x)g(x)\,dx, \tag{13}$$

where we define $g(x) := q(x)/\sqrt{p(x)}$ on $X$ and $g(x) := 0$ for $x \notin X$. Moreover, by Lemma A.8,

$$\chi^2(\mu \parallel \mu^*) + 1 = \int g(x)^2\,dx.$$

The assumption $\chi^2(\mu \parallel \mu^*) < \infty$ then implies $g \in L^2(\mathbf{R}^n)$. By Proposition A.7, we have $\phi_\epsilon * g \in L^2(\mathbf{R}^n)$. Next notice that

$$\left| \mathcal{E}_\epsilon(\mu) - \left( \chi^2(\mu \parallel \mu^*) + 1 \right) \right| \leq \int |(\phi_\epsilon * g)(x) - g(x)| g(x) \, \mathrm{d}x \leq \|\phi_\epsilon * g - g\|_2 \|g\|_2 < \infty.$$

This shows the first claim. Finally, the last expression goes to 0 as $\epsilon \to 0$ since $\|\phi_\epsilon * g - g\|_2 \to 0$ by Proposition A.7. $\qquad \square$

**Remark A.9.** *One may ask if similar results as Theorem 3.3 for $f$-divergences other than $\chi^2$ divergence can be obtained using similar techniques. We think the proof of Theorem 3.3 is highly specific to $\chi^2$ divergence because in (13) there are two copies of $q$ coming from the definition of $\mathcal{E}_\epsilon$ as a double integral over $\mu$. An $f$-divergence between $\mu$ and $\mu^*$ has the form $\int_X f(q(x)/p(x)) \, \mathrm{d}p(x)$, and the only way to have $q^2$ showing up is to choose $f(x) = x^2$ corresponding to $\chi^2$ divergence.*

**Remark A.10.** *It is possible to prove versions of Theorem 3.3 when $X$ has Hausdorff dimension $d < n$: in such cases $\chi^2$-divergence still makes sense as does (2). When $X$ is "flat", i.e., with Hausdorff dimension $d$ and contained in a $d$-dimensional linear subspace of $\mathbf{R}^n$, e.g., when $X$ is defined by a set of linear inequalities, then Theorem 3.3 follows if we adapt the assumptions of Definition 3.1 and calculation in the proof of Theorem 3.3 to be in the subspace. For a general $d$-dimensional $X$, a similar calculation yields $\mathcal{E}_\epsilon(\mu) = \int_X \left( \int_X \phi_\epsilon(x-y) \left( \frac{q}{\sqrt{p}} \right)(y) \, \mathrm{d}\mathcal{H}_d(y) \right) \left( \frac{q}{\sqrt{p}} \right)(x) \, \mathrm{d}\mathcal{H}_d(x)$. For a similar argument to go through, we will need the normalizing constant $\int_X \phi_\epsilon(x-y) \, \mathrm{d}\mathcal{H}_d(y)$ to the same for all $x$. This is true when $X$ is a $d$-dimensional sphere, but for a general $X$ the last integral will depend on the base point $x$. Proving a version of Theorem 3.3 for $X$ with Hausdorff dimension $d < n$ is an interesting future direction.*

## A.3 CONVEXITY AND $\Gamma$-CONVERGENCE

We start by recalling a few definitions regarding functionals in $\mathcal{P}(\mathbf{R}^n)$.

**Definition A.11.** *We say a functional $\mathcal{F} : \mathcal{P}(\mathbf{R}^n) \to (-\infty, \infty]$ is* proper *if there exists $\mu \in \mathcal{P}(\mathbf{R}^n)$ such that $\mathcal{F}(\mu) < \infty$, and is* lower semicontinuous (l.s.c.) *if for any weakly convergence sequence $\mu_k \to \mu$, we have $\liminf_{k \to \infty} \mathcal{F}(\mu_k) \geq \mathcal{F}(\mu)$.*

**Lemma A.12.** *For any $\epsilon > 0$, the functional $\mathcal{E}_\epsilon : \mathcal{P}(\mathbf{R}^n) \to (-\infty, \infty]$ is proper and l.s.c. Moreover, if $X$ is compact, the minimum $\min_{\mu \in \mathcal{P}(X)} \mathcal{E}_\epsilon(\mu)$ is attained by some measure in $\mathcal{P}(X)$.*

*Proof.* Taking any $x \in X$, since $\phi_\epsilon$ is bounded and $p(x) > 0$, we see that $\mathcal{E}_\epsilon(\delta_x) < \infty$ so $\mathcal{E}_\epsilon$ is proper. Moreover, given weakly convergence $\mu_k \to \mu$, by Portmanteau theorem and the fact that $W_\epsilon$ is nonnegative and l.s.c., we conclude that $\mathcal{E}_\epsilon$ is also l.s.c.

The set of probability distributions $\mathcal{P}(X) \subset \mathcal{P}(\mathbf{R}^n)$ is tight by the compactness of $X$. It is closed since if $\{\mu_k\} \subset \mathcal{P}(X)$ weakly converges to $\mu$, then by Portmanteau theorem, $\mu(X) \geq \limsup \mu(X) = 1$ so that $\mu \in \mathcal{P}(X)$. Hence by Prokhorov's theorem (Theorem 5.1.3 (Ambrosio et al., 2005)), $\mathcal{P}(X)$ is (sequentially) compact. It is then an elementary result that any l.s.c. function attains its minimum on a compact set. $\qquad \square$

We next prove Proposition 3.5 regarding the convexity of $\mathcal{E}_\epsilon$ and the uniqueness of its minimum.

*Proof of Proposition 3.5.* Let $\mu$ be any finite signed Borel measure. The compactness assumption and the fact that $p \in C^1(X)$ imply $p(x) > \delta$ for any $x \in X$ for some $\delta > 0$. Hence $p(x)^{-1/2} \leq \delta^{-1/2}$, so the weighted measure $\tilde{\mu}$ defined by $\mathrm{d}\tilde{\mu}(x) := p^{-1/2}(x) \, \mathrm{d}\mu(x)$ is also finite. By the definition of i.s.p.d. kernels, we have $\mathcal{E}_{W_\epsilon}(\mu) = \mathcal{E}_{k_\epsilon}(\tilde{\mu}) > 0$ if $\tilde{\mu}$ is not the zero measure, which is equivalent to $\mu$ not being zero since $p > 0$. Thus $W_\epsilon$ is i.s.p.d. on $X$. Also note that $(p(x)p(y))^{-1/2} < \delta^{-1}$ for all $x, y \in X$, so $\mathcal{E}_{W_\epsilon}$ is always finite. By Lemma 1.1 of Pronzato & Zhigljavsky (2021), we conclude that $\mathcal{E}_{W_\epsilon}$ is strictly convex in $\mathcal{M}_{\mathrm{sign}}$, the space of finite signed measures, and in particular it is convex on $\mathcal{P}(X)$. Hence combined with the existence result from Lemma A.12 we conclude $\mathcal{E}_\epsilon$ attains a unique minimum in $\mathcal{P}(X)$. $\qquad \square$

Our goal in the rest of this section is to prove the $\Gamma$-convergence of $\mathcal{E}_\epsilon$ to $\mathcal{E} : \mathcal{P}(\mathbf{R}^n) \to [0, \infty]$ defined in (12). By Lemma A.8, we have $\mathcal{E}(\mu) = \chi^2(\mu \parallel \mu^*) + 1$. Hence together we will have proved Theorem 3.6.

**Definition A.13** ($\Gamma$-convergence). *A sequence of functionals $\mathcal{F}_\epsilon : \mathcal{P}(\mathbf{R}^n) \to (-\infty, \infty]$ is said to $\Gamma$-converge to $\mathcal{F} : \mathcal{P}(\mathbf{R}^n) \to (-\infty, \infty]$, denoted as $\mathcal{F}_\epsilon \overset{\Gamma}{\to} \mathcal{F}$, if:*

  (a) *For any sequence $\mu_\epsilon \in \mathcal{P}(\mathbf{R}^n)$ converging weakly to $\mu \in \mathcal{P}(\mathbf{R}^n)$, $\liminf_{\epsilon \to 0} \mathcal{F}_\epsilon(\mu_\epsilon) \geq \mathcal{F}(\mu)$;*

  (b) *For any $\mu \in \mathcal{P}(\mathbf{R}^n)$, there exists a sequence $\mu_\epsilon \in \mathcal{P}(\mathbf{R}^n)$ converging weakly to $\mu$ with $\limsup_{\epsilon \to 0} \mathcal{F}_\epsilon(\mu_\epsilon) \leq \mathcal{F}(\mu)$.*

We will show $\mathcal{E}_\epsilon \overset{\Gamma}{\to} \mathcal{E}$ using Fourier transforms and Bochner's theorem.

**Definition A.14** (Fourier transform). *For $f \in L^1(\mathbf{R}^n)$, its* Fourier transform *$\hat{f}$ is the complex-valued function defined via*

$$\hat{f}(\xi) := \int e^{-2\pi i \xi \cdot x} f(x) \, dx.$$

*More generally, for a signed finite measure $\mu \in \mathcal{M}_{\mathrm{sign}}(\mathbf{R}^n)$, its* Fourier transform *$\hat{\mu}$ is the complex-valued function defined via*

$$\hat{\mu}(\xi) := \int e^{-2\pi i \xi \cdot x} \, d\mu(x).$$

*This integral is always well-defined and moreover $\hat{\mu}$ is uniformly continuous; see Borodachov et al. (2019, Section 1.10).*

We will prove the following weak version (under the additional assumption that a mollifier $\phi$ is integrable) of Bochner's theorem suitable for our case. In particular we will need the Fourier inversion formula which is not given in the usual statement of Bochner's theorem. On the other hand, we cannot directly use the Fourier inversion formula since it is not obvious how to check the integrability of $\hat{\phi}$ when $\phi$ is a mollifier.

**Lemma A.15.** *Suppose $\phi \in L^1(\mathbf{R}^n)$ is even, bounded, continuous, and $k(x, y) := \phi(x - y)$ is i.s.p.d. on any compact sets. Then its Fourier transform $\hat{\phi}$ is real, nonnegative, and the following inversion formula holds:*

$$\phi(x) = \int e^{2\pi i x \cdot \xi} \hat{\phi}(\xi) \, d\xi \quad \text{for all } x \in \mathbf{R}^n. \tag{14}$$

*Proof.* The proof is adapted from that of Varadhan (2001, Theorem 2.7) and is extended to the multi-dimensional case.

Since $\hat{\phi}(\xi) := \int e^{-2\pi i x \cdot \xi} \phi(x) \, dx$ and $\phi(x) = \phi(-x)$, with a change of variable $x' = -x$, we obtain $\overline{\hat{\phi}(\xi)} = \hat{\phi}(\xi)$, so $\hat{\phi}$ is real.

Next we show that $\hat{\phi}$ is nonnegative. For $T > 0$, we compute, for a fixed $\xi \in \mathbf{R}^n$,

$$\frac{1}{T^n} \int_{[0,T]^n} \int_{[0,T]^n} e^{-2\pi i (t-s) \cdot \xi} \phi(t - s) \, dt \, ds$$

$$= \frac{1}{T^n} \int_{[-T,T]^n} \left( \int_{\prod_i [|u_i|, 2T - |u_i|]} 2^{-n} \, dv \right) e^{-2\pi i u \cdot \xi} \phi(u) \, du$$

$$= \frac{1}{T^n} \int_{[-T,T]^n} \left( \prod_{i=1}^n \frac{2T - 2|u_i|}{2} \right) e^{-2\pi i u \cdot \xi} \phi(u) \, du$$

$$= \int_{[-T,T]^n} \left( \prod_{i=1}^n \left( 1 - \frac{|u_i|}{T} \right) \right) e^{-2\pi i u \cdot \xi} \phi(u) \, du,$$

where we have used change of variable $u = t - s$, $v = t + s$. Since $\phi \in L^1(\mathbf{R}^n)$, by dominated convergence theorem, we have as $T \to \infty$

$$\hat{\phi}(\xi) = \int e^{-2\pi i \xi \cdot x} \phi(x) \, \mathrm{d}x = \lim_{T \to \infty} \frac{1}{T^n} \int_{[0,T]^n} \int_{[0,T]^n} e^{-2\pi i (t-s) \cdot \xi} \phi(t - s) \, \mathrm{d}t \, \mathrm{d}s.$$

For $t, s \in \mathbf{R}^n$, we have

$$\mathrm{Re}\left(e^{-2\pi i (t-s) \cdot \xi} \phi(t - s))\right)$$
$$= \cos(2\pi t \cdot \xi) k(t,s) \cos(2\pi s \cdot \xi) + \sin(2\pi t \cdot \xi) k(t,s) \sin(2\pi s \cdot \xi).$$

For a fixed $T$, if we define a finite measure $\mu$ as $\mathrm{d}\mu = \mathbf{1}_{t \in [0,T]^n} \cos(2\pi t \cdot \xi) \, \mathrm{d}t$, then $k$ being i.s.p.d. implies

$$\iint \phi(t - s) \, \mathrm{d}\mu \, \mathrm{d}\mu = \int_{[0,T]^n} \int_{[0,T]^n} \cos(2\pi t \cdot \xi) \phi(t - s) \cos(2\pi s \cdot \xi) \, \mathrm{d}t \, \mathrm{d}s \geq 0,$$

and similarly for sin. Since $\hat{\phi}$ is real, we conclude that $\hat{\phi}$ is nonnegative.

Finally we prove Equation (14). For $\sigma > 0$, define $\hat{\phi}_\sigma(\xi) := \hat{\phi}(\xi) e^{-\sigma^2 \|\xi\|_2^2}$. Since $\hat{\phi}$ is bounded (because $\phi \in L^1(\mathbf{R}^n)$), we see that $\hat{\phi}_\sigma \in L^1(\mathbf{R}^n)$. We compute, for $x \in \mathbf{R}^n$, using Fubini's theorem,

$$\int e^{2\pi i x \cdot \xi} \hat{\phi}_\sigma(\xi) \, \mathrm{d}\xi = \int \hat{\phi}(\xi) e^{-\sigma^2 \|\xi\|_2^2} e^{2\pi i x \cdot \xi} \, \mathrm{d}\xi$$

$$= \int \left( \int e^{-2\pi i y \cdot \xi} \phi(y) \, \mathrm{d}y \right) e^{-\sigma^2 \|\xi\|_2^2} e^{2\pi i x \cdot \xi} \, \mathrm{d}\xi$$

$$= \int \left( \int e^{-2\pi i (y-x) \cdot \xi} e^{-\sigma^2 \|\xi\|_2^2} \, \mathrm{d}\xi \right) \phi(y) \, \mathrm{d}y$$

$$= \int (\pi/\sigma^2)^{n/2} e^{-\pi^2 \|x-y\|_2^2 / \sigma^2} \phi(y) \, \mathrm{d}y,$$

where we use the Fourier transform formula (Borodachov et al., 2019, (4.4.1)) of the Gaussian distribution. Notice that $p_\sigma(y) := (\pi/\sigma^2)^{n/2} e^{-\pi^2 \|x-y\|_2^2 / \sigma^2}$ is the density of a multivariate Gaussian centered at $x$ with covariance $\sigma^2/(2\pi^2) \cdot \mathbf{I}$. Hence $\int e^{2\pi i x \cdot \xi} \hat{\phi}_\sigma(\xi) \, \mathrm{d}\xi = (p_\sigma * \phi)(0)$. Since $\phi$ is bounded by assumption, with $x = 0$ we find $\int \hat{\phi}_\sigma(\xi) \, \mathrm{d}\xi \leq \|\phi\|_\infty$. Taking $\sigma \to 0$, by monotone convergence theorem, we have $\int \hat{\phi}(\xi) \, \mathrm{d}\xi \leq \|\phi\|_\infty$, so together with the fact that $\hat{\phi} \geq 0$ we have $\hat{\phi} \in L^1(\mathbf{R}^n)$. Finally, since $\hat{\phi}_\sigma \leq \hat{\phi}$, by dominated convergence theorem and Proposition A.4 (note $p_\sigma$ is centered at $x$), we have

$$\int e^{2\pi i x \cdot \xi} \hat{\phi}(\xi) \, \mathrm{d}\xi = \lim_{\sigma \to 0} \int e^{2\pi i x \cdot \xi} \hat{\phi}_\sigma(\xi) \, \mathrm{d}\xi = \lim_{\sigma \to 0} (p_\sigma * \phi)(0) = \phi(x).$$

$\square$

**Proposition A.16.** *Given $\phi : \mathbf{R}^n \to \mathbf{R}$ satisfying the assumptions of Lemma A.15, for any $\nu \in \mathcal{M}_{\mathrm{sign}}(\mathbf{R}^n)$, it holds that*

$$\iint \phi(x - y) \, \mathrm{d}\nu(x) \, \mathrm{d}\nu(y) = \int |\hat{\nu}(\xi)|^2 \hat{\phi}(\xi) \, \mathrm{d}\xi,$$

*Proof.* By Lemma A.15,

$$\iint \phi(x - y) \, \mathrm{d}\nu(x) \, \mathrm{d}\nu(y) = \iint \left( \int e^{-2\pi i (x-y) \cdot \xi} \hat{\phi}(\xi) \, \mathrm{d}\xi \right) \mathrm{d}\nu(x) \, \mathrm{d}\nu(y)$$

$$= \int \left( \int e^{-2\pi i x \cdot \xi} \, \mathrm{d}\nu(x) \right) \left( \int e^{2\pi i y \cdot \xi} \, \mathrm{d}\nu(y) \right) \hat{\phi}(\xi) \, \mathrm{d}\xi$$

$$= \int \hat{\nu}(\xi) \overline{\hat{\nu}(\xi)} \hat{\phi}(\xi) \, \mathrm{d}\xi = \int |\hat{\nu}(\xi)|^2 \hat{\phi}(\xi) \, \mathrm{d}\xi.$$

where we use Fubini's theorem (all measures are finite and $e^{-2\pi i \cdot}$ is bounded) to exchange the order of integration. $\square$

*Proof of Theorem 3.6.* First note that Definition A.13(b) is immediate from Theorem 3.3 with $\mu_\epsilon = \mu$: if $\mathcal{E}(\mu) = \infty$, then trivially $\limsup_{\epsilon \to 0} \mathcal{E}_\epsilon(\mu) \leq \mathcal{E}(\mu)$; otherwise we apply Theorem 3.3. So we focus on proving criterion Definition A.13(a).

Fix a sequence $\mu_\epsilon \in \mathcal{P}(\mathbf{R}^n)$ that converges weakly to $\mu \in \mathcal{P}(\mathbf{R}^n)$, and our goal is to show $\liminf_{\epsilon \to 0} \mathcal{E}_\epsilon(\mu_\epsilon) \geq \mathcal{E}(\mu)$. Without loss of generality, we may assume $\mathcal{E}_\epsilon(\mu_\epsilon) < \infty$ for all $\epsilon > 0$ (these terms have no effect in $\liminf$), which implies $\mu_\epsilon(X) = 1$. By Portmanteau's theorem and the assumption that $X$ is closed, we have $\mu(X) \geq \limsup_{\epsilon \to 0} \mu_\epsilon(X) = 1$. So all of $\mu_\epsilon$ and $\mu$ will have support in $X$.

For $m > 0, \epsilon > 0$, define a nonnegative measure $\nu_{\epsilon,m}$ by

$$\mathrm{d}\nu_{\epsilon,m}(x) := h_m(x)p^{-1/2}(x)\,\mathrm{d}\mu_\epsilon(x),$$

where $h_m : \mathbf{R}^n \to \mathbf{R}$ is a continuous monotonically decreasing cutoff function satisfying $h_m(x) = 1$ if $\|x\|_2^2 < m$ and $h_m(x) = 0$ if $\|x\|_2^2 > m + 1$, and that $h_m(x) \leq h_{m'}(x)$ for $m < m'$. Then since $p > 0$ is continuous, it is bounded below on any compact set in $X$, and hence $\nu_{\epsilon,m}$ is finite. Also define, for $m > 0$, a nonnegative measure $\nu_m$ by

$$\mathrm{d}\nu_m(x) := h_m(x)p^{-1/2}(x)\,\mathrm{d}\mu(x),$$

which is again finite following the same reasoning.

Then for any $m > 0$, denoting $\mathrm{d}\nu_\epsilon(x) := p^{-1/2}(x)\,\mathrm{d}\mu_\epsilon(x)$,

$$\mathcal{E}_\epsilon(\mu_\epsilon) = \iint \phi_\epsilon(x - y)\,\mathrm{d}\nu_\epsilon(x)\,\mathrm{d}\nu_\epsilon(y)$$
$$\geq \iint \phi_\epsilon(x - y)\,\mathrm{d}\nu_{\epsilon,m}(x)\,\mathrm{d}\nu_{\epsilon,m}(y) = \int |\hat{\nu}_{\epsilon,m}(\xi)|^2 \hat{\phi}_\epsilon(\xi)\,\mathrm{d}\xi, \tag{15}$$

where we apply Proposition A.16 for the last equality. On the other hand, note that

$$\hat{\nu}_{\epsilon,m}(\xi) = \int e^{-2\pi i \xi \cdot x}\,\mathrm{d}\nu_{\epsilon,m}(x) = \int e^{-2\pi i \xi \cdot x} h_m(x)p^{-1/2}(x)\,\mathrm{d}\mu_\epsilon(x).$$

Since $\mu_\epsilon \to \mu$ weakly and the last integrand is a continuous bounded function, we have

$$\lim_{\epsilon \to 0} \hat{\nu}_{\epsilon,m}(\xi) = \int e^{-2\pi i \xi \cdot x} h_m(x)p^{-1/2}(x)\,\mathrm{d}\mu(x) = \hat{\nu}_m(\xi).$$

On the other hand, $\hat{\phi}_\epsilon(\xi) = \int e^{-2\pi i \xi \cdot x}\phi_\epsilon(x)\,\mathrm{d}x$. Since by Definition 3.1, $\phi_\epsilon\,\mathrm{d}x$ converges to $\delta_0$ in probability (and in particular weakly), we have $\lim_{\epsilon \to 0} \hat{\phi}_\epsilon(\xi) = 1$.

Applying Fatou's lemma to (15), we obtain, for any $m > 0$,

$$\liminf_{\epsilon \to 0} \mathcal{E}_\epsilon(\mu_\epsilon) \geq \int |\hat{\nu}_m(\xi)|^2\,\mathrm{d}\xi.$$

If $\int |\hat{\nu}_m(\xi)|^2\,\mathrm{d}\xi = \infty$ for any $m > 0$, then we are done since $\liminf_{\epsilon \to 0} \mathcal{E}_\epsilon(\mu_\epsilon) = \infty$. Otherwise, by Kühn (2016, Lemma 2.11), for every $m > 0$, $\nu_m$ has density in $L^2(\mathbf{R}^n)$. This imples $\mu$ has density everywhere. Suppose $\mathrm{d}\mu(x) = q(x)\,\mathrm{d}\mathcal{L}_n(x)$. By Plancherel's theorem and the monotone convergence theorem, we have

$$\liminf_{\epsilon \to 0} \mathcal{E}_\epsilon(\mu_\epsilon) \geq \lim_{m \to \infty} \int |\hat{\nu}_m(\xi)|^2\,\mathrm{d}\xi$$
$$= \lim_{m \to \infty} \int_X \left(h_m(x)p^{-1/2}(x)q(x)\right)^2\,\mathrm{d}x$$
$$= \int_X p^{-1}(x)q(x)^2\,\mathrm{d}x = \mathcal{E}(\mu).$$

This completes the proof of $\mathcal{E}_\epsilon \overset{\Gamma}{\to} \mathcal{E}$.

Now suppose $X$ is compact, and let $\mu_\epsilon^* := \arg\min_{\mu \in \mathcal{P}(X)} \mathcal{E}_\epsilon(\mu)$. To establish $\mu_\epsilon^* \to \mu^*$ weakly (resp. $\lim_{\epsilon \to 0} \mathcal{E}_\epsilon(\mu_\epsilon^*) = \mathcal{E}(\mu^*) = 1$), it suffices to show that every subsequence of $\{\mu_\epsilon^*\}$ (resp.

$\{\mathcal{E}_\epsilon(\mu_\epsilon^*)\}$) has a further convergence subsequence converging to $\mu^*$ (resp. $\mathcal{E}(\mu^*)$). With a slight abuse of notation, we assume the sequence of $\epsilon$ is chosen so that $\{\mu_\epsilon^*\}$ (resp. $\{\mathcal{E}_\epsilon(\mu_\epsilon^*)\}$) is already some subsequence of the original sequence. As argued in the proof of Lemma A.12, $\mathcal{P}(X)$ is compact with respect to weak convergence. Hence $\{\mu_\epsilon^*\}$ has a weakly convergence subsequence $\mu_{\epsilon_k}^* \to \nu$ for some $\nu \in \mathcal{P}(X)$. The $\Gamma$-convergence $\mathcal{E}_\epsilon \xrightarrow{\Gamma} \mathcal{E}$ implies

$$\liminf_{k\to\infty} \mathcal{E}_{\epsilon_k}(\mu^*) \geq \liminf_{k\to\infty} \mathcal{E}_{\epsilon_k}(\mu_{\epsilon_k}^*) \geq \mathcal{E}(\nu) \geq \limsup_{k\to\infty} \mathcal{E}_{\epsilon_k}(\nu) \geq \limsup_{k\to\infty} \mathcal{E}_{\epsilon_k}(\mu_{\epsilon_k}^*),$$

where we have used, for each inequality, $\mu_{\epsilon_k}^* = \arg\min_{\mathcal{P}(X)} \mathcal{E}_{\epsilon_k}$, Definition A.13(a), the first paragraph of this proof, and again the fact that $\mu_{\epsilon_k}^* = \arg\min_{\mathcal{P}(X)} \mathcal{E}_{\epsilon_k}$. Since $\liminf_{k\to\infty} \mathcal{E}_{\epsilon_k}(\mu^*) = \mathcal{E}(\mu^*)$ by Theorem 3.3, we have

$$\mathcal{E}(\mu^*) \geq \lim_{k\to\infty} \mathcal{E}_{\epsilon_k}(\mu_{\epsilon_k}^*) = \mathcal{E}(\nu) \geq \mathcal{E}(\mu^*),$$

where the last inequality follows because $\mu^*$ is the minimizer of $\mathcal{E}$. Then $\lim_{k\to\infty} \mathcal{E}_{\epsilon_k}(\mu_{\epsilon_k}^*) = \mathcal{E}(\mu^*) = 1$. Moreover, $\chi^2(\nu \parallel \mu^*) = 0$. This can only happen if $\nu$ and $\mu^*$ agree on all Borel sets, so $\nu = \mu^*$. $\qquad\square$

## A.4 DIFFERENTIAL CALCULUS OF $\mathcal{E}_\epsilon$ IN $\mathcal{P}_2(\mathbf{R}^n)$

**Lemma A.17.** *Let $x_0 \in \mathbf{R}$, $h > 0$, and $\Omega \subset \mathbf{R}^m$ be a compact set. Suppose $f : (x_0 - h, x_0 + h) \times \Omega \to \mathbf{R}$ is jointly continuous and the derivative $\frac{\partial}{\partial x} f : (x_0 - h, x_0 + h) \times \Omega \to \mathbf{R}$ exists and is jointly continuous. Then $\int_\Omega f(x, \omega)\, \mathrm{d}\omega$ is differentiable for $x \in (x_0 - h, x_0 + h)$, and*

$$\frac{\mathrm{d}}{\mathrm{d}x} \int_\Omega f(x, \omega)\, \mathrm{d}\omega = \int_\Omega \frac{\partial}{\partial x} f(x, \omega)\, \mathrm{d}\omega$$

*where the integration is with respect to the Lebesgue measure $\mathcal{L}_m$ on $\Omega$.*

*Proof.* Fix $x \in (x_0 - h, x_0 + h)$ and let $t > 0$ be small enough such that $[x - t, x + t] \subset (x_0 - h, x_0 + h)$. Note that $\int_\Omega f(x, \omega)\, \mathrm{d}\omega$ is well-defined by the dominated convergence theorem since $\sup_{x \in [x-t, x+t], \omega \in \Omega} f(x, \omega) < \infty$ and $\mathcal{L}_m(\Omega) < \infty$. Define $\theta(\omega) := \sup_{x \in [x-t, x+t]} \left| \frac{\partial}{\partial x} f(x, \omega) \right|$ which is finite since $\frac{\partial}{\partial x} f(x, \omega)$ is continuous, so that $\theta(\omega) \geq \frac{\partial}{\partial x} |f(x, \omega)|$ for all $x \in [x - t, x + t]$ and $\theta$ is integrable since $\mathcal{L}_m(\Omega) < \infty$. Hence by the differentiation lemma (Klenke, 2013, Theorem 6.28) we are done. $\qquad\square$

The following lemma is similar to Korba et al. (2021, Proposition 3) but with different assumptions: we do not put any integrability assumptions on $W_\epsilon$, but we do restrict measures to have compact support. For a differentiable $f : \mathbf{R}^n \times \mathbf{R}^n \to \mathbf{R}$, we use $\nabla_1 f(x, y)$ to denote the gradient with respect to $x$. We also use $\mathrm{H}_1 f(x, y)$ to denote the Hessian of $f$ with respect to $x$. We similarly denote $\nabla_2 f(x, y)$ and $\mathrm{H}_2 f(x, y)$.

**Lemma A.18.** *Assume $\mu \in \mathcal{P}_2(\mathbf{R}^n)$ has density $q \in C_c^1(\mathbf{R}^n)$. Let $\boldsymbol{\xi} \in C(\mathbf{R}^n; \mathbf{R}^n)$. Denote $\boldsymbol{s}(x, t) := x + t\boldsymbol{\xi}(x)$. Then for all $t > 0$ and all $\epsilon > 0$,*

$$\frac{\mathrm{d}}{\mathrm{d}t} \mathcal{E}_\epsilon(\mu_t) = 2 \iint \left( \boldsymbol{\xi}(x)^\top \nabla_1 W_\epsilon(\boldsymbol{s}(x, t), \boldsymbol{s}(y, t)) \right) \mathrm{d}\mu(x)\, \mathrm{d}\mu(y). \tag{16}$$

*In particularly,*

$$\frac{\mathrm{d}}{\mathrm{d}t}\bigg|_{t=0} \mathcal{E}_\epsilon(\mu_t) = 2 \iint \left( \boldsymbol{\xi}(x)^\top \nabla_1 W_\epsilon(x, y) \right) \mathrm{d}\mu(x)\, \mathrm{d}\mu(y). \tag{17}$$

*Moreover,*

$$\frac{\mathrm{d}^2}{\mathrm{d}t^2}\bigg|_{t=0} \mathcal{E}_\epsilon(\mu_t) = 2 \iint \left( \boldsymbol{\xi}(x)^\top \nabla_1 \nabla_2 W_\epsilon(x, y) \boldsymbol{\xi}(y) + \boldsymbol{\xi}(x)^\top \mathrm{H}_1 W_\epsilon(x, y) \boldsymbol{\xi}(x) \right) \mathrm{d}\mu(x)\, \mathrm{d}\mu(y). \tag{18}$$

*Proof.* We compute

$$\frac{\mathrm{d}}{\mathrm{d}t}\mathcal{E}_\epsilon(\mu_t) = \frac{\mathrm{d}}{\mathrm{d}t}\left(2\iint W_\epsilon(\boldsymbol{s}(x,t),\boldsymbol{s}(y,t))\,\mathrm{d}\mu(x)\,\mathrm{d}\mu(y)\right).$$

Since $\mu$ has compact support and $(x,y,t) \mapsto W_\epsilon(\boldsymbol{s}(x,t),\boldsymbol{s}(y,t))q(x)q(y)$ is jointly continuous and its derivative with respect to $t$ is also jointly continuous, by Lemma A.17, we can push $\frac{\mathrm{d}}{\mathrm{d}t}$ inside the double integral and we obtain (16). Another application of Lemma A.17 shows that if we take derivative with respect to $t$ again on (16) and evaluate at 0 we obtain (18). □

**Lemma A.19.** *Let $f \in C^1(\mathbf{R}^n)$, $p \in [1,\infty]$. Assume $\phi_\epsilon$ has compact support for some $\epsilon > 0$. Then for all $i = 1, \ldots, n$, $\phi_\epsilon * f$ is differentiable and*

$$\frac{\partial}{\partial x_i}(\phi_\epsilon * f)(x) = \left(\phi_\epsilon * \frac{\partial}{\partial x_i}f\right)(x).$$

*Proof.* Since $\mathrm{supp}(\phi_\epsilon)$ is compact and $f \in C^1(\mathbf{R}^n)$ is bounded on any compact set, we know $\phi_\epsilon * f$ is well-defined at every $x \in \mathbf{R}^n$. Note that

$$\frac{\partial}{\partial x_i}(\phi_\epsilon * f)(x) = \frac{\partial}{\partial x_i}\left(\int f(x-y)\phi_\epsilon(y)\,\mathrm{d}y\right)$$

$$\overset{(?)}{=} \int \frac{\partial f}{\partial x_i}(x-y)\phi_\epsilon(y)\,\mathrm{d}y = \left(\phi_\epsilon * \frac{\partial}{\partial x_i}f\right)(x).$$

Since $\mathrm{supp}(\phi_\epsilon)$ is compact and $(x,y) \mapsto f(x-y)\phi(y)$ is $C^1$, by Lemma A.17 we justify the existence of the derivative and the exchange of differentiation and integration (?). □

### A.4.1 SUBDIFFERENTIALS OF $\mathcal{E}_\epsilon$

Recall the following notion of a "Wasserstein gradient" in $\mathcal{P}_2(\mathbf{R}^n)$ from Ambrosio et al. (2005, Definition 10.1.1).

**Definition A.20.** *A vector field $\boldsymbol{w} \in L^2(\mu;\mathbf{R}^n)$ is a strong Fréchet subdifferential of a functional $\mathcal{F} : \mathcal{P}_2(\mathbf{R}^n) \to (-\infty,+\infty]$ if for all $\boldsymbol{T} \in L^2(\mu;\mathbf{R}^n)$, the following holds:*

$$\mathcal{F}(\boldsymbol{T}_{\#}\mu) - \mathcal{F}(\mu) \geq \int \boldsymbol{w}(x)^\top(\boldsymbol{T}(x) - x)\,\mathrm{d}\mu(x) + o\left(\|\boldsymbol{T} - \boldsymbol{I}\|_{L^2(\mu;\mathbf{R}^n)}\right). \qquad (19)$$

Note that we cannot apply Ambrosio et al. (2005, Lemma 10.4.1) directly to prove (4) because interaction energies cannot be written in the form of (10.4.1) in their setup.

*Proof of Proposition 3.7.* Let $\boldsymbol{\xi} \in C_c^\infty(\mathbf{R}^n;\mathbf{R}^n)$. By Lemma A.18, we have

$$\frac{\mathrm{d}}{\mathrm{d}t}\bigg|_{t=0}\mathcal{E}_\epsilon(\mu_t) = 2\iint \left(\boldsymbol{\xi}(x)^\top \nabla_1 W_\epsilon(x,y)\right)\mathrm{d}\mu(x)\,\mathrm{d}\mu(y)$$

$$= 2\iint \left(\boldsymbol{\xi}(x)^\top \nabla_1\left(\phi_\epsilon(x-y)(p(x)p(y))^{-1/2}\right)\right)q(y)\,\mathrm{d}y\,\mathrm{d}\mu(x)$$

$$= 2\int \boldsymbol{\xi}(x)^\top \nabla\left(p(x)^{-1/2}(\phi_\epsilon * q/\sqrt{p})(x)\right)\mathrm{d}\mu(x),$$

where the last step follows from applying Lemma A.17 since $q$ has compact support. Now suppose $\boldsymbol{w} \in L^2(\mu;\mathbf{R}^n)$ is a strong Fréchet subdifferential satisfying (19). For the sequence $\{\boldsymbol{T}_t\}$, we have by definition

$$\mathcal{E}_\epsilon(\mu_t) - \mathcal{E}_\epsilon(\mu) \geq \int \boldsymbol{w}(x)^\top(\boldsymbol{T}_t(x) - x)\,\mathrm{d}\mu(x) + o\left(\|\boldsymbol{T}_t - \boldsymbol{I}\|_{L^2(\mu;\mathbf{R}^n)}\right)$$

$$= \int \boldsymbol{w}(x)^\top(t\boldsymbol{\xi}(x))\,\mathrm{d}\mu(x) + o(t).$$

Hence

$$\liminf_{t\downarrow 0}\frac{\mathcal{E}_\epsilon(\mu_t) - \mathcal{E}_\epsilon(\mu)}{t} \geq \int \boldsymbol{w}(x)^\top \boldsymbol{\xi}(x)\,\mathrm{d}\mu(x) \geq \liminf_{t\uparrow 0}\frac{\mathcal{E}_\epsilon(\mu_t) - \mathcal{E}_\epsilon(\mu)}{t}.$$

The previous calculation shows $\frac{\mathrm{d}}{\mathrm{d}t}|_{t=0}\mathcal{E}_\epsilon(\mu_t)$ exists, and hence it is equal to $\int \boldsymbol{w}(x)^\top \boldsymbol{\xi}(x)\,\mathrm{d}\mu(x)$. This proves for any $\boldsymbol{\xi} \in C_c^\infty(\mathbf{R}^n; \mathbf{R}^n)$,

$$\int \boldsymbol{w}(x)^\top \boldsymbol{\xi}(x)\,\mathrm{d}\mu(x) = \int \boldsymbol{\xi}(x)^\top \nabla\left(2p(x)^{-1/2}(\phi_\epsilon * q/\sqrt{p})(x)\right)\mathrm{d}\mu(x).$$

Hence we have shown (4).

Finally, we show $\lim_{\epsilon \to 0} \boldsymbol{w}_\epsilon(x) = \boldsymbol{w}_{\chi^2}(x)$ for $\mu$-a.e. $x$ under the additional assumption that $\phi_\epsilon$ has compact support. By Ambrosio et al. (2005, Lemma 10.4.1) with $F(x, \rho(x)) = \left(\frac{\rho(x)}{p(x)} - 1\right)^2 p(x)$, we find the strong subdifferential of $\chi^2(\cdot \parallel \mu^*)$ is given by

$$\boldsymbol{w}_{\chi^2,\mu}(x) = 2\nabla\frac{q(x)}{p(x)}, \text{ for } \mu\text{-a.e. } x \in \mathbf{R}^n. \tag{20}$$

To show (5), we compute, for $\mu$-a.e. $x$,

$$\begin{aligned}
\boldsymbol{w}_\epsilon(x) &= \nabla\left(p(x)^{-1/2}(\phi_\epsilon * q/\sqrt{p})(x)\right) \\
&= \nabla(p(x)^{-1/2})(\phi_\epsilon * q/\sqrt{p})(x) + p(x)^{-1/2}\nabla(\phi_\epsilon * q/\sqrt{p})(x) \\
&= \nabla(p(x)^{-1/2})(\phi_\epsilon * q/\sqrt{p})(x) + p(x)^{-1/2}(\phi_\epsilon * \nabla(q/\sqrt{p}))(x),
\end{aligned}$$

where for the last equality we have applied Lemma A.19. Now taking $\epsilon \to 0$, by Proposition A.4 using the fact that $\mathrm{supp}(\phi_\epsilon)$ is compact (so that $q/\sqrt{p}$ and $\nabla(q/\sqrt{p})$ are bounded on $x + \mathrm{supp}(\phi_\epsilon)$), we obtain (5). $\qquad\square$

### A.4.2 DISPLACEMENT CONVEXITY OF $\mathcal{E}_\epsilon$ AT $\mu^*$ AS $\epsilon \to 0$

The statement of Proposition 3.8 is similar in form as Korba et al. (2021, Corollary 4) but in our case we do not have the second term in Equation (18) vanishing and we need to take the limit $\epsilon \to 0$. We also do not resort to RKHS theory in the proof.

*Proof of Proposition 3.8.* By (18), we have

$$\frac{\mathrm{d}^2}{\mathrm{d}t^2}\bigg|_{t=0}\mathcal{E}_\epsilon(\mu_t) = 2(\mathcal{F}_\epsilon + \mathcal{G}_\epsilon),$$

where

$$F_\epsilon = \iint \left(\boldsymbol{\xi}(x)^\top \nabla_1 \nabla_2 W_\epsilon(x, y)\boldsymbol{\xi}(y)\right)\mathrm{d}\mu^*(x)\,\mathrm{d}\mu^*(y)$$

$$G_\epsilon = \iint \left(\boldsymbol{\xi}(x)^\top \mathrm{H}_1 W_\epsilon(x, y)\boldsymbol{\xi}(x)\right)\mathrm{d}\mu^*(x)\,\mathrm{d}\mu^*(y).$$

We tackle $F_\epsilon$ first. Observe that successive application of integration by parts using the fact that $\boldsymbol{\xi}$ has compact support gives

$$\begin{aligned}
F_\epsilon &= \sum_{i,j=1}^n \iint \boldsymbol{\xi}_i(x)\frac{\partial^2}{\partial x_i \partial y_j}W_\epsilon(x, y)\boldsymbol{\xi}_j(y)p(x)p(y)\,\mathrm{d}x\,\mathrm{d}y \\
&= -\sum_{i,j=1}^n \iint \frac{\partial}{\partial x_i}(\boldsymbol{\xi}_i(x)p(x))\frac{\partial}{\partial y_j}W_\epsilon(x, y)\boldsymbol{\xi}_j(y)p(y)\,\mathrm{d}x\,\mathrm{d}y \\
&= \sum_{i,j=1}^n \iint \frac{\partial}{\partial x_i}(\boldsymbol{\xi}_i(x)p(x))W_\epsilon(x, y)\frac{\partial}{\partial y_j}(\boldsymbol{\xi}_j(y)p(y))\,\mathrm{d}x\,\mathrm{d}y \\
&= \iint \left(\sum_{i=1}^n \frac{\partial}{\partial x_i}(\boldsymbol{\xi}_i(x)p(x))\right)W_\epsilon(x, y)\left(\sum_{j=1}^n \frac{\partial}{\partial y_j}(\boldsymbol{\xi}_j(y)p(y))\right)\mathrm{d}x\,\mathrm{d}y.
\end{aligned}$$

If we view $\sum_{i=1}^n \frac{\partial}{\partial x_i}(\boldsymbol{\xi}_i(x)p(x))$ as the density of a signed measure (it is integrable since it has compact support), and since $W_\epsilon$ is i.s.p.d. on the support of $\boldsymbol{\xi}$ by Proposition 3.5(a), we see that each double integral in the last expression is non-negative. Hence $F_\epsilon \geq 0$.

Next we show $\lim_{\epsilon \to 0} G_\epsilon = 0$. Since $\mu^*$ has compact support, by Fubini's theorem,

$$G_\epsilon = \int \boldsymbol{\xi}(x)^\top \mathrm{H}_1 \left( \int W_\epsilon(x,y)p(y)\,\mathrm{d}y \right) \boldsymbol{\xi}(x)p(x)\,\mathrm{d}x.$$

To expand the integral inside the Hessian operator, we have

$$\mathrm{H}_1 \left( \int W_\epsilon(x,y)p(y)\,\mathrm{d}y \right) = \mathrm{H}\left( p(x)^{-1/2}(\phi_\epsilon * \sqrt{p})(x) \right).$$

First by the chain rule and Lemma A.19, we have

$$\frac{\mathrm{d}}{\mathrm{d}x}\left( p(x)^{-1/2}(\phi_\epsilon * \sqrt{p})(x) \right) = \frac{\mathrm{d}}{\mathrm{d}x}\left( p(x)^{-1/2} \right)(\phi_\epsilon * \sqrt{p}) + p(x)^{-1/2}\left( \phi_\epsilon * \frac{\mathrm{d}}{\mathrm{d}x}\sqrt{p}(x) \right).$$

Differentiating again while applying Lemma A.19, we obtain after rearranging terms,

$$\begin{aligned}
\mathrm{H}_1 \left( \int W_\epsilon(x,y)p(y)\,\mathrm{d}y \right) = & \left( \frac{\mathrm{d}^2}{\mathrm{d}x^2}p(x)^{-1/2} \right)(\phi_\epsilon * \sqrt{p})(x) \\
& + 2\left( \frac{\mathrm{d}}{\mathrm{d}x}p(x)^{-1/2} \right)\left( \phi_\epsilon * \frac{\mathrm{d}}{\mathrm{d}x}\sqrt{p} \right)(x) \\
& + p(x)^{-1/2}\left( \phi_\epsilon * \frac{\mathrm{d}^2}{\mathrm{d}x^2}\sqrt{p} \right)(x).
\end{aligned} \quad (21)$$

By Proposition A.4 and the fact that $p \in C_c^2(\mathbf{R}^n)$, we have

$$\begin{aligned}
& \lim_{\epsilon \to 0} \mathrm{H}_1 \left( \int W_\epsilon(x,y)p(y)\,\mathrm{d}y \right) \\
= & \left( \frac{\mathrm{d}^2}{\mathrm{d}x^2}p(x)^{-1/2} \right)\sqrt{p}(x) + 2\left( \frac{\mathrm{d}}{\mathrm{d}x}p(x)^{-1/2} \right)\frac{\mathrm{d}}{\mathrm{d}x}\sqrt{p}(x) + p(x)^{-1/2}\frac{\mathrm{d}^2}{\mathrm{d}x^2}\sqrt{p}(x) \\
= & \frac{\mathrm{d}^2}{\mathrm{d}x^2}\left( p(x)^{-1/2}\sqrt{p(x)} \right) = 0.
\end{aligned}$$

Finally, we have

$$\begin{aligned}
\lim_{\epsilon \to 0} G_\epsilon &= \lim_{\epsilon \to 0} \int \boldsymbol{\xi}(x)^\top \mathrm{H}_1 \left( \int W_\epsilon(x,y)p(y)\,\mathrm{d}y \right) \boldsymbol{\xi}(x)p(x)\,\mathrm{d}x \\
&= \int \boldsymbol{\xi}(x)^\top \left( \lim_{\epsilon \to 0} \mathrm{H}_1 \left( \int W_\epsilon(x,y)p(y)\,\mathrm{d}y \right) \right) \boldsymbol{\xi}(x)p(x)\,\mathrm{d}x \\
&= 0,
\end{aligned}$$

where interchanging the limit and the integral is jusftified by the dominated convergence theorem and the fact that $p$ and $\phi_\epsilon$ have compact support (we need compact support assumption of $\phi_\epsilon$ to make sure convolutions appearing in (21) are uniformly bounded when $\epsilon$ is sufficiently small).

$\square$

### A.4.3 A DESCENT LEMMA FOR $\mathcal{E}_\epsilon$ WITH TIME DISCRETIZATION

We now consider a time discretization of the gradient flow of $\mathcal{E}_\epsilon$ given by, for an initial measure $\mu_0 \in \mathcal{P}_2(\mathbf{R}^n)$ and $m \in \mathbf{Z}_{>0}$, with a step size $\gamma > 0$,

$$\mu_{m+1} := (\boldsymbol{I} - \gamma \boldsymbol{w}_{\epsilon,\mu_m})_\# \mu_m. \quad (22)$$

Similar to Korba et al. (2021, Proposition 14), we can show the following descent lemma for iterations (22) with the following additional assumptions:

**Assumption A.21.** *Suppose the target density* $p \in C^2(\mathbf{R}^n)$ *satisfies* $1/C_p \leq p(x) \leq C_p$, $\|p(x)\|_2 \leq C'_p$, $\|\mathrm{H}p(x)\|_2 \leq C''_p$ *for some* $C_p, C'_p, C''_p$ *for all* $x \in \mathbf{R}^n$, *where we use* $\|A\|_2$ *to indicate the matrix spectral norm (i.e.* $\|A\|_2 = \sigma_{\max}(A)$). *Suppose the mollifier* $\phi_\epsilon \in C^2(\mathbf{R}^n)$ *satisfies, in addition to Definition 3.1,* $\phi_\epsilon(x) \leq C_\epsilon$, $\|\nabla\phi_\epsilon(x)\|_2 \leq C'_\epsilon$, *and* $\|\mathrm{H}\phi_\epsilon(x)\|_2 \leq C''_\epsilon$ *for some* $C_\epsilon, C'_\epsilon, C''_\epsilon$.

**Lemma A.22.** *Under Assumption A.21, there exists a constant* $L > 0$ *such that the function* $\nabla_1 W_\epsilon :$ $\mathbf{R}^n \times \mathbf{R}^n \to \mathbf{R}^n$ *is* $L$-*Lipschitz in terms of* $\|\cdot\|_2$ *in either input.*

*Proof.* Denote $r(x) := p(x)^{-1/2}$. Then our assumptions imply $r(x) \leq C_r := C_p^{1/2}$, $\|\nabla r(x)\|_2 \leq C'_r := \frac{1}{2}C_p^{3/2}C'_p$, and $\|\mathrm{H}_1 r(x)\|_2 \leq C''_r := \frac{3}{4}C_p^{5/2}C'^2_p + \frac{1}{2}C_p^{3/2}C''_p$. We compute, for $x, y \in \mathbf{R}^n$,

$$\nabla_1 W_\epsilon(x,y) = \nabla_x \left(\phi_\epsilon(x-y)r(x)r(y)\right)$$
$$= \nabla\phi_\epsilon(x-y)r(x)r(y) + \phi_\epsilon(x-y)\nabla r(x)r(y).$$

Then

$$\mathrm{H}_1 W_\epsilon(x,y) = \mathrm{H}\phi_\epsilon(x-y)r(x)r(y) + 2\nabla\phi_\epsilon(x-y)\nabla r(x)^\top r(y) + \phi_\epsilon(x-y)\mathrm{H}r(x)r(y),$$

and

$$\nabla_2 \nabla_1 W_\epsilon(x,y) = -\mathrm{H}\phi_\epsilon(x-y)r(x)r(y) + \nabla\phi_\epsilon(x-y)r(x)\nabla r(y)^\top$$
$$- \nabla\phi_\epsilon(x-y)\nabla r(x)r(y) + \phi_\epsilon(x-y)\nabla r(x)\nabla r(y)^\top.$$

Then we have

$$\|\mathrm{H}_1 W_\epsilon(x,y)\|_2 \leq C''_\epsilon C_r^2 + 2C'_\epsilon C'_r C_r + C_\epsilon C''_r C_r,$$
$$\|\nabla_2 \nabla_1 W_\epsilon(x,y)\|_2 \leq C''_\epsilon C_r^2 + 2C'_\epsilon C_r C'_r + C_\epsilon C'^2_r.$$

Hence we conclude that $\nabla_1 W_\epsilon$ is $L$-Lipschitz with

$$L := C''_\epsilon C_r^2 + 2C'_\epsilon C'_r C_r + C_\epsilon \max(C''_r C_r, C'^2_r).$$

$\square$

**Remark A.23.** *Compared with Korba et al. (2021, Lemma 1), to ensure* $\nabla_1 W_\epsilon$ *is Lipscthiz, we only require uniform boundedness of* $p$ *and* $\phi_\epsilon$ *up to second order derivatives instead of up to order 3.*

**Proposition A.24.** *Under Assumption A.21, suppose* $\mu_0 \in \mathcal{P}_2(\mathbf{R}^n)$ *has compact support. Then for any* $\gamma \leq \frac{1}{2L}$, *with* $L$ *defined as in Lemma A.22 and* $\{\mu_m\}_{m>0}$ *defined as in (22),*

$$\mathcal{E}_\epsilon(\mu_{m+1}) - \mathcal{E}_\epsilon(\mu_m) \leq -\gamma(1-2\gamma L)\|\boldsymbol{w}_{\epsilon,\mu_m}\|_{L^2(\mu_m)} \leq 0. \tag{23}$$

*Proof.* We first show $\mu_m$ has compact support for all $m \in \mathbf{Z}_{\geq 0}$. For $\mu \in \mathcal{P}_2(\mathbf{R}^n)$ with compact support, the proof of Proposition 3.7 implies (note Proposition 3.7 assumes $\mu$ has density but it is not necessary to obtain the following formula using the same proof)

$$\boldsymbol{w}_{\epsilon,\mu}(x) = 2\int \nabla_1 W_\epsilon(x,y)\,\mathrm{d}\mu(y).$$

Since $x \mapsto \nabla_1 W_\epsilon(x,y)$ is $C^1$ by the assumptions and $\mu$ has compact support, by Lemma A.17, we see that $\boldsymbol{w}_{\epsilon,\mu}$ is differentiable with

$$\nabla\boldsymbol{w}_{\epsilon,\mu}(x) = 2\int \mathrm{H}_1 W_\epsilon(x,y)\,\mathrm{d}\mu(y).$$

By induction, suppose $\mu_m$ has compact support. Then

$$\mathrm{supp}(\mu_{m+1}) \subset (\boldsymbol{I} - \gamma\boldsymbol{w}_{\epsilon,\mu_m})_\# \mathrm{supp}(\mu_m) \subset \mathrm{supp}(\mu_m) + \gamma\left(\sup_{x \in \mathrm{supp}(\mu_m)}\|\boldsymbol{w}_{\epsilon,\mu_m}\|_2\right)B_1(0).$$

Since $\boldsymbol{w}_{\epsilon,\mu_m}$ is continuous, the set on the right-hand side is bounded. Hence $\mu_{m+1}$ has compact support.

Now fix $m \in \mathbf{Z}_{>0}$ and we show (23). Define a path $\{\mu_t\}_{t \in [0,1]}$ defined by $\mu_t = (\boldsymbol{I} - \gamma t \boldsymbol{w}_{\epsilon,\mu_m})_{\#}\mu_m$. Let $f(t) := \mathcal{E}_\epsilon(\mu_t)$. By Lemma A.18, the continuity of $\boldsymbol{w}_{\epsilon,\mu_m}$ implies that $f$ is differentiable. Moreover, another application of Lemma A.17 implies that $f$ is twice differentiable, and in particular continuously differentiable. Hence $f$ is absolutely continuous on the compact interval $[0,1]$. By the fundamental theorem of calculus, we have

$$\mathcal{E}_\epsilon(\mu_{m+1}) - \mathcal{E}_\epsilon(\mu_m) = f(1) - f(0) = f'(0) + \int_0^1 (f'(t) - f'(0))\,\mathrm{d}t.$$

Observe that by Lemma A.18,

$$
\begin{aligned}
f'(0) &= 2 \iint (-\gamma \boldsymbol{w}_{\epsilon,\mu_m}(x))^\top \nabla_1 W_\epsilon(x,y)\,\mathrm{d}\mu_m(x)\,\mathrm{d}\mu_m(y) \\
&= \int (-\gamma \boldsymbol{w}_{\epsilon,\mu_m}(x)) \left(2 \int \nabla_1 W_\epsilon(x,y)\,\mathrm{d}\mu_m(y)\right) \mathrm{d}\mu_m(x) \\
&= -\gamma \|\boldsymbol{w}_{\epsilon,\mu_m}(x)\|_{L^2(\mu_m)}^2,
\end{aligned}
$$

where we apply Fubini's theorem to exchange the order of integration together with the fact that $\mu_m$ has compact support and the integrand is continuous. Let $\boldsymbol{s}(x,t) := x - \gamma t \boldsymbol{w}_{\epsilon,\mu_m}(x)$. Note that, again by Lemma A.18,

$$
\begin{aligned}
|f'(t) - f'(0)| &\leq 2 \iint \left|(-\gamma \boldsymbol{w}_{\epsilon,\mu_m}(x))^\top \left(\nabla_1 W_\epsilon(\boldsymbol{s}(x,t),\boldsymbol{s}(y,t)) - \nabla_1 W_\epsilon(x,y)\right)\right| \mathrm{d}\mu_m(x)\,\mathrm{d}\mu_m(y) \\
&\leq 2\gamma \iint \|\boldsymbol{w}_{\epsilon,\mu_m}(x)\|_2 \|\nabla_1 W_\epsilon(\boldsymbol{s}(x,t),\boldsymbol{s}(y,t)) - \nabla_1 W_\epsilon(x,y)\|_2\,\mathrm{d}\mu_m(x)\,\mathrm{d}\mu_m(y).
\end{aligned}
$$

Note that, by Lemma A.22, we have

$$
\begin{aligned}
&\|\nabla_1 W_\epsilon(\boldsymbol{s}(x,t),\boldsymbol{s}(y,t)) - \nabla_1 W_\epsilon(x,y)\|_2 \\
\leq{}& \|\nabla_1 W_\epsilon(\boldsymbol{s}(x,t),\boldsymbol{s}(y,t)) - \nabla_1 W_\epsilon(\boldsymbol{s}(x,t),y)\|_2 + \|\nabla_1 W_\epsilon(\boldsymbol{s}(x,t),y) - \nabla_1 W_\epsilon(x,y)\|_2 \\
\leq{}& L\gamma t(\|\boldsymbol{w}_{\epsilon,\mu_m}(x)\|_2 + \|\boldsymbol{w}_{\epsilon,\mu_m}(y)\|_2).
\end{aligned}
$$

Thus

$$
\begin{aligned}
|f'(t) - f'(0)| &\leq 2\gamma^2 Lt \iint \|\boldsymbol{w}_{\epsilon,\mu_m}(x)\|_2 \left(\|\boldsymbol{w}_{\epsilon,\mu_m}(x)\|_2 + \|\boldsymbol{w}_{\epsilon,\mu_m}(y)\|_2\right) \mathrm{d}\mu_m(x)\,\mathrm{d}\mu_m(y) \\
&\leq 2\gamma^2 Lt \left(\|\boldsymbol{w}_{\epsilon,\mu_m}\|_{L^2(\mu_m)} + \left(\int \|\boldsymbol{w}_{\epsilon,\mu_m}(x)\|_2\,\mathrm{d}\mu_m(x)\right)^2\right) \\
&\leq 4\gamma^2 Lt \|\boldsymbol{w}_{\epsilon,\mu_m}\|_{L^2(\mu_m)},
\end{aligned}
$$

where we have used the Cauchy-Schwartz inequality in the last step.

Combining everything, we have shown

$$
\begin{aligned}
\mathcal{E}_\epsilon(\mu_{m+1}) - \mathcal{E}_\epsilon(\mu_m) &= f'(0) + \int_0^1 (f'(t) - f'(0))\,\mathrm{d}t \\
&\leq -\gamma(1 - 2\gamma L)\|\boldsymbol{w}_{\epsilon,\mu_m}\|_{L^2(\mu_m)} \leq 0,
\end{aligned}
$$

since $\gamma < \frac{1}{2L}$.

$\square$

# B  WEIGHTED HYPERSINGULAR RIESZ ENERGY

We recall results most relevant to us from Borodachov et al. (2019). Suppose $X \subset \mathbf{R}^n$ is compact, of Hausdorff dimension $d$, and $d$-rectifiable, i.e., the image of a bounded set in $\mathbf{R}^d$ under a Lipschitz mapping. Given a non-vanishing continuous probability density $p$ on $X$, define a measure $\mathcal{H}_d^p(B) :=$

$\int_B p(x)\,\mathrm{d}\mathcal{H}_d(x)$ for any Borel set $B \subset X$. The target measure $h_d^p$ is defined to be $h_d^p(B) := \mathcal{H}_d^p(B)/\mathcal{H}_d^p(X)$. The *weighted $s$-Riesz energy* is defined as the interaction energy

$$E_s(\omega_N) := \sum_{i \neq j} \frac{(p(x_i)p(x_j))^{-s/2d}}{\|x_i - x_j\|^s}. \tag{24}$$

The following result states that minimizers of the weighted $s$-Riesz energy approximate the target measure when the $s$ is sufficiently large.

**Theorem B.1** (Theorem 11.1.2, Borodachov et al. (2019)). *Suppose $s > d$. For $\omega_N^* \in \arg\min E_s$ with $\omega_N^* = \{x_1^N, \ldots, x_N^N\}$, we have weak convergence $\delta_{\omega_N^*} \to h_d^p$ as $N \to \infty$.*

A similar result with a slightly different assumption holds for $s = d$ (Theorem 11.1.3 of Borodachov et al. (2019)).

**Comparison of** (24) **with discretized MIE** (6). Our theory is only valid for the full-dimensional case, i.e., $n = d$ (see a few exceptions discussed in Remark A.10). When this is the case and if the mollifier is taken to be from the $s$-Riesz family, (6) becomes, for $s > n$,

$$E_\epsilon(\omega_N) := \sum_{i,j=1}^N \frac{(p(x_i)p(x_j))^{-1/2}}{\left(\|x_i - x_j\|_2^2 + \epsilon^2\right)^{s/2}}.$$

Compared with (24), in our case there is an $\epsilon$ in the denominator, so that the continuous version of the energy $\mathcal{E}_\epsilon$ does not blow up all the time (this is in contrast with the continuous version of (24)—see Borodachov et al. (2019, Theorem 4.3.1)). Moreover, the exponential scaling on $p$ is different: in our case we use $-1/2$ whereas in (24) it is $-s/2n$. The diagonal $i = j$ is included in our energy, but this is inconsequential as another valid discretization is to discard the diagonal term (Borodachov et al., 2019, Theorem 4.2.2). On the other hand, our theory allows a bigger class of mollifiers that are not necessarily of Riesz families. We also allow $X$ to be non-compact. An interesting future research direction is to extend our theory to cases where $d < n$, e.g., when $X$ is an embedded $d$-dimensional submanifold in $\mathbf{R}^n$.

## C   Algorithmic details

In this section we provide algorithmic details of MIED and compare with the updates of SVGD (Liu & Wang, 2016). The negative gradient of (7) with respect to $x_i$ is

$$-\nabla_{x_i} \log E_\epsilon(\omega_N) = 2\sum_{j \neq i} \frac{e^{I_{ij}}}{\sum_{i,j} e^{I_{ij}}} \left((\nabla \log \phi_\epsilon)(x_j - x_i) + \frac{1}{2}\nabla \log p(x_i)\right)$$
$$+ \frac{e^{I_{ii}}}{\sum_{i,j} e^{I_{ij}}} \nabla \log p(x_i)$$
$$= \sum_{j=1}^N \frac{e^{I_{ij}}}{\sum_{i,j} e^{I_{ij}}} \left(2\nabla \log \phi_\epsilon(x_j - x_i) + \nabla \log p(x_i)\right),$$

where

$$I_{ij} := \log \phi_\epsilon(x_i - x_j) - \frac{1}{2}(\log p(x_i) + \log p(x_j)), \tag{25}$$

and to get the last equality we used the fact that $\nabla \phi(0) = 0$ thanks to the assumption $\phi(x) = \phi(-x)$. Then gradient descent on (7) gives our algorithm (Algorithm 1). The special treatment of the diagonal terms described in Section 4 amounts to modifying only the diagonal $I_{ii}$.

---

**Algorithm 1:** Mollified interaction energy descent (MIED) in the logarithmic domain.

---

**Input:** target density $p$, mollifier $\phi_\epsilon$, initial particles $\{x_i^0\}_{i=1}^N$, learning rate $\eta$, total steps $T$.

**for** $t \leftarrow 1$ **to** $T$ **do**

    **for** $i \leftarrow 1$ **to** $N$ **do**

        $x_i^{t+1} \leftarrow x_i^t + \eta \sum_{j=1}^N \frac{e^{I_{ij}}}{\sum_{i,j} e^{I_{ij}}} \left(2\nabla \log \phi_\epsilon(x_j - x_i) + \nabla \log p(x_i)\right)$, where $I_{ij}$ is

        defined in (25);

    **end**

**end**

**return** final particles $\{x_i^T\}_{i=1}^N$.

---

## C.1    COMPARISON WITH SVGD

The update formula in Algorithm 1 is similar to the one in SVGD: if we use $\phi_\epsilon(x - y)$ in place of the kernel $k(x, y)$ in the SVGD update and rewrite:

$$x_i^{t+1} = x_i^t + \eta \sum_{j=1}^N \left(\nabla \phi_\epsilon(x_j - x_i) + \phi_\epsilon(x_j - x_i)\nabla \log p(x_j)\right)$$

$$= x_i^t + \eta \sum_{j=1}^N \phi_\epsilon(x_j - x_i) \left(\nabla \log \phi_\epsilon(x_j - x_i) + \nabla \log p(x_j)\right). \quad \text{(SVGD)}$$

For both algorithms, the update formula for each particle consists of attraction and repulsion terms and the total time complexity of each update iteration is $O(N^2)$. We note the following differences. First, in our formulation we have scaling factors $\frac{e^{I_{ij}}}{\sum_{i,j} e^{I_{ij}}}$ which help stabilizing the optimization (as a by-product of working in the logarithmic domain) and put more weight on nearby particles as well as particles in low-density regions, whereas in (SVGD) the scaling factors are $\phi_\epsilon(x_j - x_i)$ which are not adapted to prioritize low-density regions. Second, in MIED, the attraction force for particle $i$ only comes from $\nabla \log p(x_i)$, whereas in (SVGD) the attraction force comes from $\nabla \log p(x_j)$ for all $j$'s. Third, for each $j$, in our formulation the repulsive force has an additional factor of 2 in front of $\nabla \log \phi_\epsilon(x_j - x_i)$.

Empirically, the additional scaling factors in MIED help produce samples with good separations compared to SVGD, since closer pairs of points will have large weights. Additionally, since MIED optimizes a finite-dimensional objective (7), we can employ accelerated gradient-based optimizers like Adam (Kingma & Ba, 2014), which we used in our experiments. In contrast, SVGD does not optimize any finite-dimensional objective. While practical SVGD implementations also use optimizers like Adam, it is unclear how the resulting particle dynamics is related to the gradient flow of KL divergence.

## C.2    HANDLING CONSTRAINTS WITH DYNAMIC BARRIER METHOD

The dynamic barrier method is introduced in Gong & Liu (2021) which solves $\min_x f(x)$ subject to $g(x) \leq 0$ where $g$ is scalar-valued. Intuitively, their method computes update directions by either decreasing $g(x)$ when $g(x) > 0$, following $-\nabla f(x)$ if the constraints are satisfied, or balancing both gradient directions.

In order to handle multiple constraints such as in Figure 11, we consider a generalized version of their dynamic barrier method. In this generalized setting, $g : \mathbf{R}^n \to \mathbf{R}^m$ is vector-valued and constraints are $g(x) \leq 0$ where the $\leq$ sign is interpreted coordinate wise.. Suppose we are at iteration $t$ with current solution $x^t$. Then the next update direction $v^*$ is taken to be the $\arg\min$ of

$$\min_{v \in \mathbf{R}^n} \left\|v - \nabla f(x^t)\right\|_2^2 \text{ s.t. } \forall i = 1, \ldots, m, \nabla g_i(x^t)^\top v \geq \alpha_i g_i(x^t), \quad (26)$$

where $\alpha_i > 0$ are fixed hyperparameters; in our implementation we simply choose $\alpha_i = 1$. Then $x^{t+1} = x^t - \eta v^*$ with learning rate $\eta$. In our implementation we use Adam (Kingma & Ba, 2014) that modulates the update directions. Observe that the optimization problem (26) is the

same as projecting a point $\nabla f(x^t)$ onto the polyhedron formed by the intersection of the halfspaces $\cap_{i=1}^m \{x \in \mathbf{R}^n : \nabla g_i(x^t)^\top v \geq \alpha_i g_i(x^t)\}$. To solve (26), we use Dykstra's algorithm (Tibshirani, 2017) which can be interpreted as running coordinate descent on the dual of (26). We use a fixed number of 20 iterations for the Dykstra's algorithm which we found to be sufficient for our experiments; in the case of a single constraint, we only need to use one iteration.

## D EXPERIMENT DETAILS AND ADDITIONAL RESULTS

### D.1 GAUSSIANS IN VARYING DIMENSIONS

We generate the $n$-dimensional Gaussians used to produce Figure 1 as follows. We generate a matrix $\sqrt{A} \in \mathbf{R}^{n \times n}$ with i.i.d. entries uniformly in $[-1, 1]$. Then we set $A = \sqrt{A}\sqrt{A}^\top / \det\left(\sqrt{A}\right)$. This way $\det(A) = 1$. We then use $A$ as the covariance matrix for the Gaussian (centered at 0). We use Adam with learning rate 0.01 for all methods for a total of 2000 iterations. This is enough for SVGD and MIED to converge, while for KSDD the convergence can be much slower.

We visualize the samples from each method for $n = 2$ in Figure 5. We notice that MIED is capable of generating well-separated samples, while for SVGD there is a gap between the inner cluster of samples and a sparse outer ring. For KSDD we see the artifact where too many samples concentrate on the diagonal.

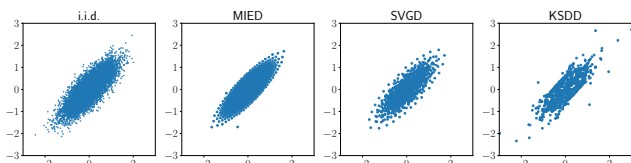

Figure 5: Visualization of samples from each method for a 2D skewed Gaussian.

### D.2 COLLAPSED SAMPLES WHEN THE KERNEL WIDTH IS TOO BIG

In Figure 3, we see that samples from SVGD collapse with an adaptive kernel where the variance is taken to be half of the median of the squared distance among all pairs of points (Liu & Wang, 2016); at termination the median of the squared distance is greater than 1 in that experiment. Here we investigate this issue further. In Figure 6, for the same uniform sampling setup, we visualize the results of SVGD with a fixed kernel width and MIED with Gaussian mollifiers with the same kernel width: when the kernel width (i.e. $2\epsilon^2$ in $\phi_\epsilon^g(x) = \exp\left(-\frac{\|x\|_2^2}{2\epsilon^2}\right)/Z_\epsilon^g$) is too big, both SVGD and MIED result in collapsed samples. This is because since the target density is a constant, the only force in the updates is the repulsive force. When the kernel width is too large, the repulsive force coming from points in the same collapsed cluster is dominated by the repulsive force coming from points from other clusters—this is evident in the update directions shown in the leftmost column of Figure 6 (SVGD with kernel width 0.1). When using the dynamic barrier method Gong & Liu (2021) to enforce the square constraints instead of reparameterization with $\tanh$, we obtain similar results as in Figure 6.

This pathological phenomenon is not only limited to constrained sampling: when sampling the 2D Gaussian from Figure 5, using too big a kernel width can also result in collapsing (Figure 7).

We emphasize that our theory of MIED suggests that in practice we need to choose the kernel width very small in order to sample from the correct target measure according to Theorem 3.3 and Theorem 3.6. In comparison, the theory of SVGD has no such implication.

### D.3 UNIFORM SAMPLING WITH AN ALTERNATIVE MIRROR MAP

In this section, we show that for sampling from a uniform distribution in the square $[-1, 1]^2$, the results of SVMD/MSVGD (Shi et al., 2021) depend heavily on the choice of the mirror map. Instead of the entropic mirror map used to produce results in Figure 3, here we use the mirror map

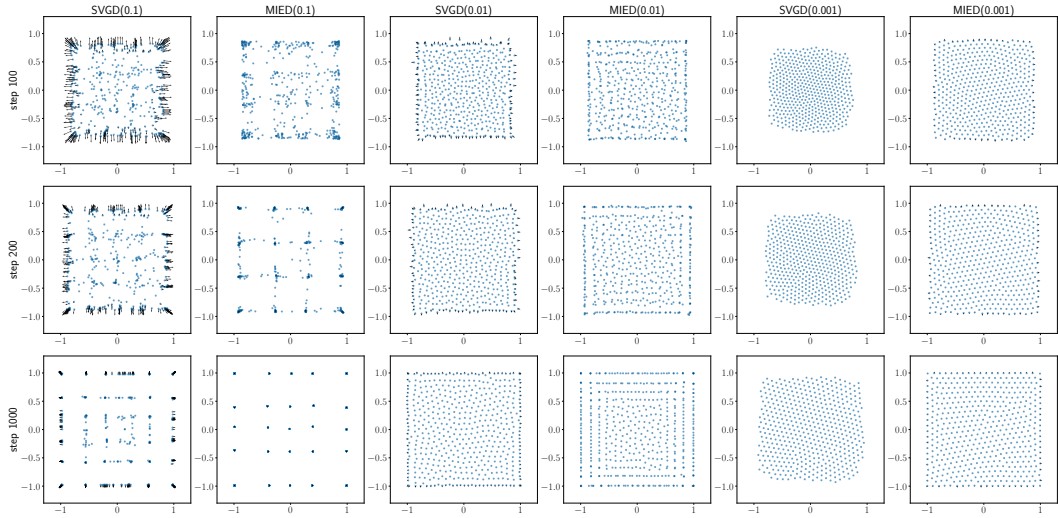

Figure 6: SVGD and MIED with fixed-size Gaussian kernels for uniform sampling in a square. In each cell of the grid, we plot the samples along with the update direction (black arrows) at that iteration. Rows correspond to iterations $100, 200, 1000$. Columns correspond to each method with varying kernel widths (twice the variance of the Gaussian kernel/mollifier) indicated in the parentheses.

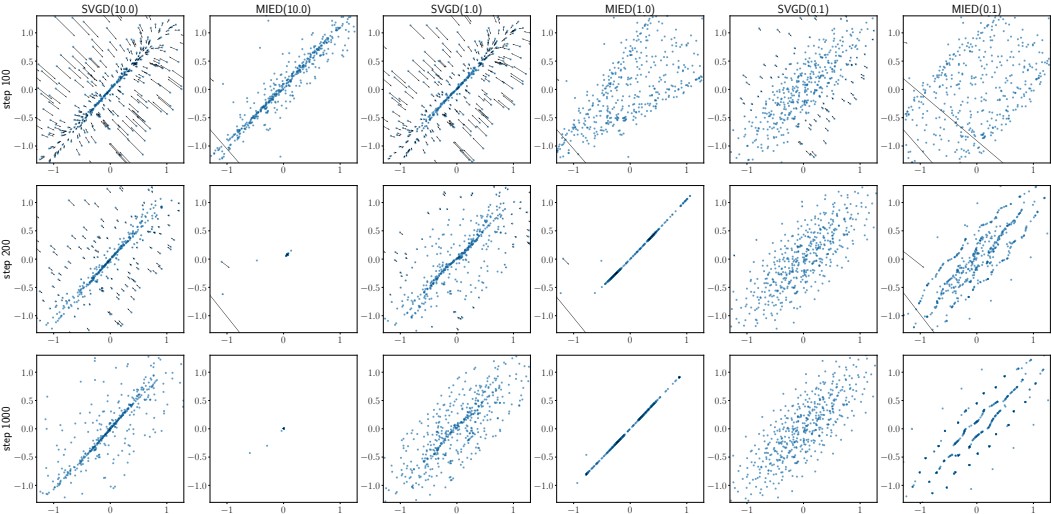

Figure 7: SVGD and MIED with fixed-size Gaussian kernels for sampling a 2D Gaussian as in Figure 5. We see that using too big a kernel size can lead to collapsed samples for both SVGD and MIED.

$\phi(\theta) = \sum_{i=1}^{n} \left( \log \frac{1}{1-\theta_i} + \log \frac{1}{1+\theta_i} \right)$ as in Ahn & Chewi (2021). The results are shown in Figure 8. SVMD/MSVGD fail to draw samples near the boundary; we suspect this is because the gradient of the conjugate $\nabla \phi^*(\eta) = (\sqrt{1+\eta^2}-1)/\eta$ (coordinate-wise arithmetic) requires coordinates of $\eta$ to go to $\infty$ to land near the boundary. We verify this phenomenon by using $\nabla \phi^*(\eta)$ as the reparametrization map in MIED (rightmost figure in Figure 8): indeed with such reparameterization MIED also struggles near the boundary.

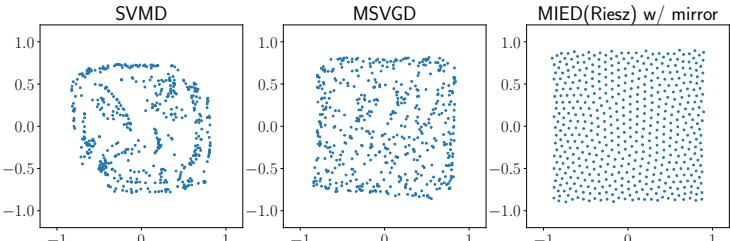

Figure 8: Visualization of samples for uniform sampling from a 2D box when using a suboptimal mirror map. All three methods fail to draw samples near the boundary of the box $[-1, 1]^2$.

### D.4 20-DIMENSIONAL DIRICHLET DISTRIBUTION

We sample from the 20-dimensional Dirichlet distribution in the same setup as in Shi et al. (2021) with 50 particles. Results and visualization are shown in Figure 9. We see that unlike sampling from a box (Figure 3), both MSVGD and SVMD by Shi et al. (2021) perform well in this experiment. This is due to the fact that the entropic mirror map used here is a well-tested choice for simplex constraint, yet obtaining a good mirror map for a generic constraint, even if it is linear, can be challenging. Our method does not have such a limitation, as it can easily incorporate existing constrained optimization tools.

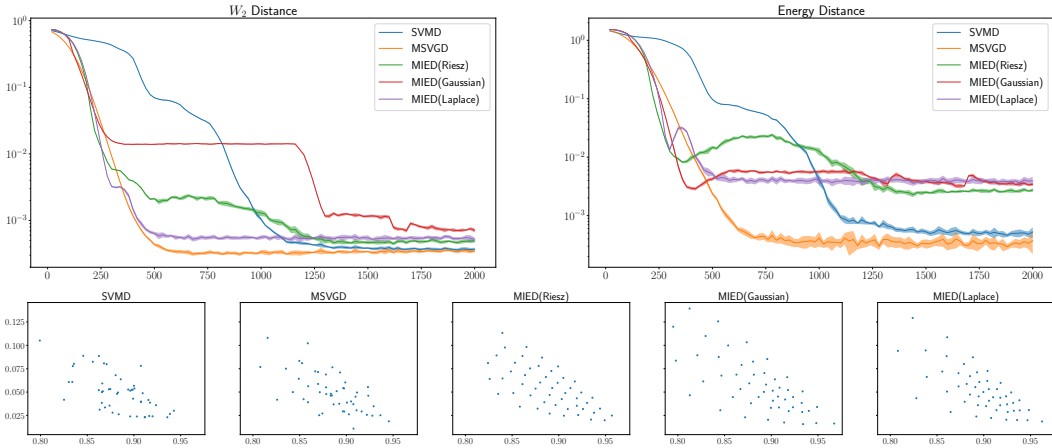

Figure 9: Top: Metrics vs. the number of iteration for sampling the 20-dimensional Dirichlet distribution. Bottom: visualization of samples from each method.

### D.5 EFFECT OF $s$ FOR RIESZ MOLLIFIERS

When we use the $s$-Riesz families of mollifiers in MIED, we have the freedom of choosing the hyperparameter $s$ so long as $s > n$. In this section, we study the effect of $s$ on the resulting samples. We consider the problem of sampling from a mixture of four 2D Gaussians centered at $(\pm 1, \pm 1)$, each with diagonal variance $0.3$ and constrained to the $[-1, 1]^2$ box. We vary $s$ in $[2, 10]$ and the number of particles $N$ in $\{100, 200, 500, 1000, 2000\}$. All runs use a total of $1000$ iterations with learning rate $0.01$. In the top of Figure 10, we plot the $W_2$ distance and energy distance as functions of $s$ for each $N$. Interestingly, we see the best performing $s$ is in $[3, 5.0]$ and depends on $N$. This suggests that our choice of $s = n + 10^{-4}$ in Section 5 may not be optimal and there is room for hyperparameter tuning to further improve the performance of MIED with Riesz kernel. At the bottom of Figure 10 we visualize the samples of MIED with $s = 3$ and of SVGD with kernel width $0.01$ (adaptive kernels would result in collapsed samples and other widths we tested on would result

in worse samples). While SVGD samples form visible artifacts, the samples of MIED are evenly distributed and the four modes of the mixtures emerge as $N$ increases.

Figure 10: Ablation study of the hyperparameter $s$ for MIED with $s$-Riesz families of mollifiers. Top: metrics as functions of $s$. Bottom: visualization of samples of MIED with a Riesz mollifier with $s = 3$ and of SVGD with kernel width $0.01$.

## D.6 MORE CONSTRAINED SAMPLING EXPERIMENTS

In this section we test MIED on more low dimensional constrained sampling problems and qualitatively assess the results. Note that mirror LMC (Zhang et al., 2020; Ahn & Chewi, 2021) or mirror SVGD (Shi et al., 2021) cannot be applied due to non-convexity of the constraints. In Figure 11, we consider uniform sampling of a challenging 2D region with initial samples drawn from the top-right corner: as the number of iterations increases, MIED gradually propagate samples to fill up the entire region. In Figure 12, we consider sampling from a von Mises-Fisher distribution on a unit sphere. Although our theory focuses on sampling from a full-dimensional distribution, as discussed in Remark A.10, we can extend Theorem 3.3 to the case of a sphere due to its symmetry. We see the samples visualized in Figure 12 capture the two modes that emerge by restricting the Gaussian to a unit sphere.

## D.7 DETAILS ON FAIRNESS BAYESIAN NEURAL NETWORK EXPERIMENT

We use $80\%/20\%$ training/test split as in Liu et al. (2021). We use the source code provided by Liu et al. (2021) with default hyperparameters. The source code provided by the authors of Liu et al. (2021) does not implement the calculation of $g(\theta)$ faithfully as written in the formula, so we corrected it. All methods use 2000 iterations for training. For our method we use learning rate $0.001$. One of their four methods (Control+SVGD) got stuck at initialization (with accuracy around $0.75$), so we omit its result from the plot in Figure 4.

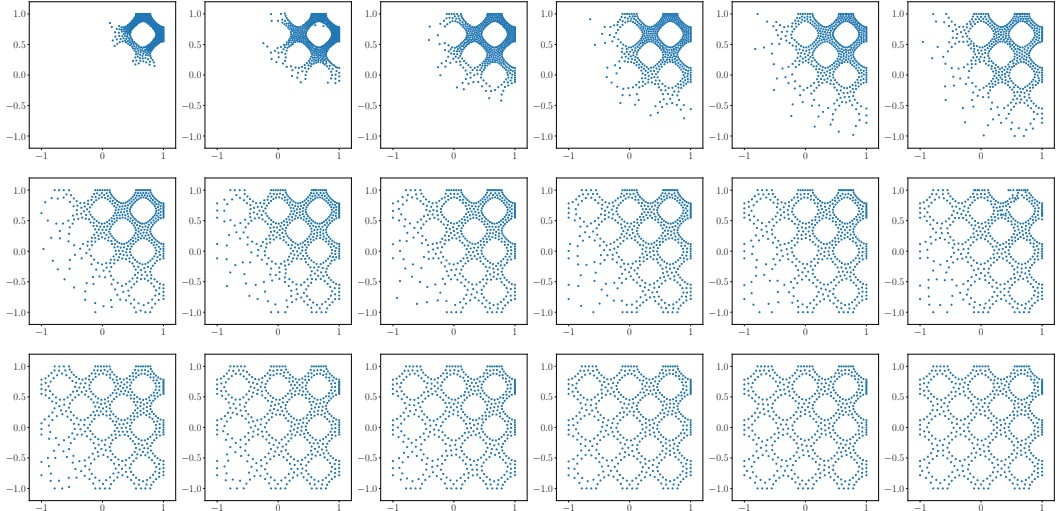

Figure 11: Uniform sampling of the region $\{(x, y) \in [-1, 1]^2 : (\cos(3\pi x) + \cos(3\pi y))^2 < 0.3\}$ using MIED with a Riesz mollifier ($s = 3$) where the constraint is enforced using the dynamic barrier method. The plot in row $i$ column $j$ shows the samples at iteration $100 + 200(6i + j)$. The initial samples are drawn uniformly from the top-right square $[0.5, 1.0]^2$.

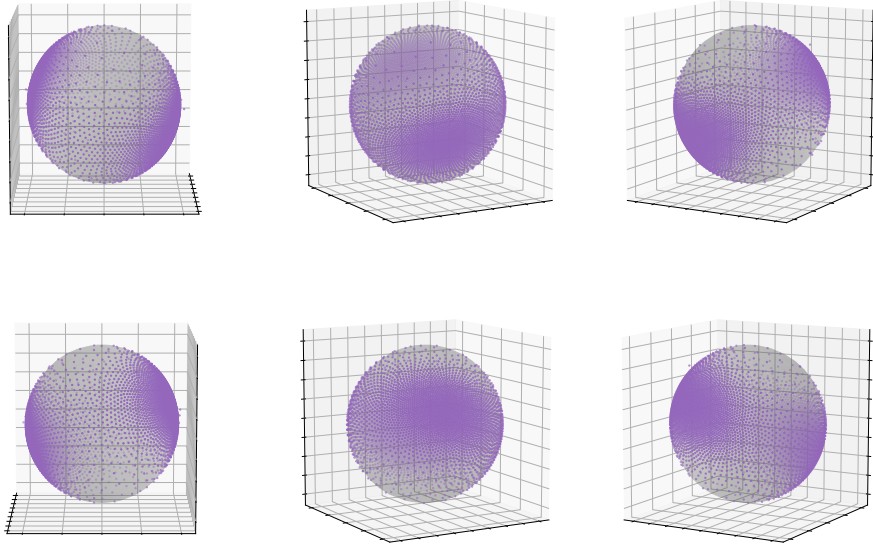

Figure 12: Sampling from the von Mises-Fisher distribution obtained by constraining the 3-dimensional Gaussian from Appendix D.1 to the unit sphere. The unit-sphere constraint is enforced using the dynamic barrier method and the shown results are obtained using MIED with Riesz kernel and $s = 3$. The six plots are views from six evenly spaced azimuthal angles.

