# OpenReview forum: "Sampling with Mollified Interaction Energy Descent"
_ICLR.cc/2023/Conference — ICLR 2023 poster_

### Official Review · Reviewer_gruH · 2022-10-21

**Confidence:** 3
**Correctness:** 4
**Technical Novelty And Significance:** 3
**Empirical Novelty And Significance:** 3
**Recommendation:** 8

**Clarity, Quality, Novelty And Reproducibility:**

The paper is very clear and nice to follow. Although it is challenging at times, I think the authors did a good job in balancing the main text and the appendix. I am not an expert in this particular field so I cannot say if they missed a paper that would really be relevant, but I feel they did their best to provide a good account for the state of the art.

Regarding reproducibility, I must say that the proposed algorithm looks so easy to implement that I don't think it is a real issue. However, the authors also do provide their code.

**Strength And Weaknesses:**

Strengths of the paper include:
* The method is well grounded theoretically and although I could not follow the proofs in detail, I trust the authors on this
* The paper is a very good balance between motivations, background, a clear explanation of the method with all the hard maths in appendix, and a very nice part about how to implement the idea in practice
* I personally found the paper very inspiring and it leaves me with many questions / ideas that I will just write down later here.
* The method seems to work well and to offer a lot of ways of improvement.

On the downsides:
* Since the method seems to work just as well as all others (see the experiments) on many aspects, I am wondering what would be the interesting features ? I must say that I like it that a paper is not bragging about how better it is everywhere, but I would appreciate it if the authors could at least provide some hints regarding when it is / is not appropriate. What about computing time ?
* I am not used to this kind of experiments and I don't exactly know if such low dimensional problems may be considered state of the art. If so, it's all right. But my core concern is whether a particle-based system would really be able to scale to high dimensions, since it would maybe mean an extreme large number of particles ? How would that work ?

**Summary Of The Paper:**

The paper proposes a new method to sample from a target distribution that is known up to a normalizing constant.
In essence, the proposed algorithm updates a set of particles by gradient descent and the loss is a smart pick that asses the energy of the pairwise interactions between the particles. When it's small, the particles follow the target distribution by design.
As I see it, the way to do it is to "smooth" the target distribution through some kernel with a bandwidth \eps, for which the problem is tractable, and then see that the smaller the bandwidth get, the close we get to the target. This "smoothing" operation is called "mollifying".

All in all, the paper is nice to read, provides a lot of useful pointers and the proposed method looks surprisingly easy to implement.

**Summary Of The Review:**

As a short summary, I think the authors do propose a principled and nice method to sample from a target distribution whose density is known up to a constant, that looks like an interesting alternative to MCMC. The paper is very inspiring to me, and leaves me with some comments.

Since I don't exactly know where to write them, here I list them below in no particular order
* 2/ pairwise interaction: you should maybe mention that $p$ is still the target here, or maybe it's obvious
* looks to me you forgot |f(x)|^p instead of f(x)^p in the notation for ||f||_p
* definition 3.2 looks super central. However, I was disapointed there was so few discussion about it, at least to provide some "feel" or rationale for it. It actually looked counter intuitive until I understood you were about to _minimize_ it, which was not obvious to me at first. Then I had to check appendix A.2 to see it in action.
* As some food for thought and just to know your opinion on this (in this thread, not necessarily in the paper), I must ask whether you think you may come up with a way to define that kind of energy for any particular divergence between distributions ? Here we have chi^2, but it looks like maybe you may find some trick to generalize your idea to any other one, or at least ones with some features to be identified.
* How restrictive is this assumption about ispd ? I am not used to it, and I wonder what it means in practice that may limit my choice of "mollifier"
* you really don't want to write just kernel ? The historical reference is nice but do we need another name ? Is it used in some community ?
* In equation (7), do we need N^2 terms ? Isn't there some symmetry that allows us using only N^2/2 ?
* if N is too large, could you consider mini batches of pairs ?
* the algorithm is nice, but do we know when to stop ? Do we have some clue whether when the samples actually match the target distribution ?
* The "special treatment of the diagonal terms" is definitely not the most elegant part of the paper. Do you think you could handle the diagonal and off-diagonal terms in a different way, maybe by balancing them in some way ?
* In this paragraph: "which can dominant", "for mollifiers we consider" -> "for the mollifiers we consider"
* I have no clue about the "dynamic barrier method". For the paper to be self contained, it could help at least describing it a bit.
* page 9 "the mirror map has to chosen" -> to be chosen
* conclusion: "the target mesaure" -> measure

---

> ### Author Response · Authors · 2022-11-14
> **Response (1/2)**
>
> Thank you for your positive review. We are glad that you like our method and the theory we developed. Please find below our response to each of your questions/comments.
>
> **Since the method seems to work just as well as all others (see the experiments) on many aspects, I am wondering what would be the interesting features. I would appreciate it if the authors could at least provide some hints regarding when it is/is not appropriate**
>
> Regarding unconstrained sampling, MIED consistently results in lower Wasserstein distances compared to SVGD and KSDD (Figure 1, 2). This is coherent with the observation that MIED generates more evenly-spaced samples and it implies that MIED will result in lower Monte Carlo errors compared to SVGD/KSDD.
>
> Regarding constrained sampling, MIED performs a lot better and is more flexible with constraint handling (Figure 3, 4). To further illustrate this point, we have conducted three more constrained sampling experiments:
> In Section D.4 we study different choices of $s$ for MIED with Riesz mollifiers on a constrained sampling problem with four mixtures of Gaussians.
> In Section D.5, Figure 10, we run MIED to sample uniformly on a challenging constrained set with localized initialization.
> In Section D.5, Figure 11, we sample a von Mises-Fisher distribution using MIED.
>
> See also our response to reviewer FquR where we compare LMC with MIED.
>
> One limitation of MIED is the bias introduced in the mollifier parameter $\epsilon$, in which case the minimizer of MIE approximates the true target density. See the second last point in our response to reviewer oXmc.
>
>
> **What about computing time?**
>
> Please refer to the last point in our response to reviewer oXmc.
>
> **But my core concern is whether a particle-based system would really be able to scale to high dimensions since it would maybe mean an extremely large number of particles. How would that work?**
>
> Please refer to the first point of our response to reviewer oXmc. We emphasize that two Bayesian experiments (one on unconstrained logistic regression, another one on constrained fairness Bayesian neural networks) are done in quite large dimensions. In particular, in the fairness experiment, we sample from dimension 4452.
>
> **Definition 3.2 looks super central. However, I was [disappointed] there was so [little] discussion about it, at least to provide some "feel" or rationale for it.**
>
> We discussed briefly after Definition 3.2 about the intuition of minimizing MIE. In the revised Section C, we explain more intuition for MIED with particles and compare it with SVGD.
>
> **...whether you think you may come up with a way to define that kind of energy for any particular divergence between distributions? Here we have $\chi^2$, but it looks like maybe you may find some trick to generalize your idea to any other one, or at least ones with some features to be identified.**
>
> This is a really good question. We have added a new Remark A.9 to answer your question. In short, the technique involved in the proof of Theorem 3.3 only works for $\chi^2$ divergence. It is possible that one can come up with a new energy with a different kind of interaction (not necessarily pairwise) and prove an analogous theorem.
>
> **How restrictive is this assumption about ispd?**
>
> This is not a restrictive assumption. The three families of mollifiers satisfy such an assumption. This is a common assumption in RKHS theory which SVGD and KSDD are based on. I.s.p.d. kernels can be constructed from any *completely monotone* function: see Example 1.1 in [Pronzato & Zhigljavsky 2021]. I.s.p.d. kernels appear in many other places. For instance, in [1], it is shown that i.s.p.d. kernels are *characteristic*, meaning the corresponding integral probability metric is indeed a metric.
>
> [1] Sriperumbudur, Bharath K., et al. "Hilbert space embeddings and metrics on probability measures." The Journal of Machine Learning Research 11 (2010): 1517-1561.

---

> > ### Comment · Reviewer_gruH · 2022-11-28
> > **thank you for your answers**
> >
> > detailed feedback and answers, thank you for this. I maintain my good score.
> >
> > I understand that from some perspective, dimension 4452 can be considered high, but from an image processing or signal processing viewpoint where dimensions are around millions, it remains quite small, so I wonder about how it would scale to that kind of problems.
> >
> > Maybe by maintaining and training some shared "autoencoder" in parallel to reduce the sampling dimension ?
> >
> >
> > in any case,  I think this is a good paper that should be accepted.

---

> > > ### Author Response · Authors · 2022-12-08
> > > **Thank you**
> > >
> > > Dear Reviewer,
> > >
> > > Thank you for your maintaining a positive evaluation of our work.
> > >
> > > The idea of sampling a latent code and then mapping it to a higher dimensional point is very interesting and could potentially scale up our method to domains like high-resolution images. We will investigate this idea in future work. A challenge would be how to evaluate samples in such high dimensions; as mentioned in our response to Reviewer oXmc, metrics such as Wasserstein distance becomes unreliable.

---

> ### Author Response · Authors · 2022-11-14
> **Response (2/2)**
>
> **You really don't want to write just kernel? The historical reference is nice but do we need another name? Is it used in some communities?**
>
> We avoid writing kernel because it does not emphasize the fact that we want its bandwidth $\epsilon$ to go to 0. In many particle-based methods such as SVGD and KSDD, kernels are used but their bandwidths are fixed constants away from 0. “Mollifiers” is a common term in PDE theory.
>
> **In equation (7), do we need N^2 terms?**
>
> You’re right. The symmetry allows using only $N^2/2$ of the terms. But this might not necessarily bring 2x speed up since we parallelize the algorithm on the GPU. We agree there is a lot of potential to improve the runtime of the algorithm; currently, as discussed at the end of Section C.1, the time complexity of MIED is on par with SVGD.
>
> **If N is too large, could you consider mini-batches of pairs ?**
>
> This is a great suggestion. We think one straightforward way to handle millions of particles is to update only a mini-batch at a time. Such a strategy has been proposed for SVGD. Studying how to scale up our method is a very interesting future direction.
>
> **The algorithm is nice, but do we know when to stop? Do we have some clue whether when the samples actually match the target distribution?**
>
> The fact that our algorithm is solving a finite-dimensional optimization problem means that we can use any stopping criterion from gradient-based optimization, for instance, stopping when the improvement of the objective is below a threshold. In contrast, for methods like SVGD and MSVGD, it is unclear when to stop because they are not optimizing any finite-dimensional objective. To test whether samples match the target distribution (for which we have density access), we can compute for example KL or chi-squared divergence but we need to estimate density from samples which can be inaccurate even in moderate dimensions.
>
> **Do you think you could handle the diagonal and off-diagonal terms in a different way, maybe by balancing them in some way?**
>
> We agree our current way of handling the diagonal terms can be improved using more sophisticated heuristics like dynamic balancing. For the experiments that we considered, the simple heuristic we used seems to work well.
>
> **I have no clue about the "dynamic barrier method". For the paper to be [self-contained], it could help at least [describe] it a bit.**
>
> We have added a new Section C.2 to describe the dynamic barrier method used in our implementation.

---

### Official Review · Reviewer_oXmc · 2022-10-25

**Confidence:** 2
**Correctness:** 4
**Technical Novelty And Significance:** 3
**Empirical Novelty And Significance:** 2
**Recommendation:** 6

**Clarity, Quality, Novelty And Reproducibility:**

The idea is reasonably clearly explained though, as I mentioned the significance and novelty of their theoretical findings is not well explained.
The algorithm is novel and sound. The experimental results (even though toy-ish and 2D) suggested that compared to the baselines such as SVGD, the particles are better spread throughout (e.g. the 2D uniform).
Also the time-complexity analysis is missing.
I have not checked the code but it seems that the results are reproducible.

Question: in page 3: $\int_X f(x)^p dx < 0$ ? or >0?

**Details Of Ethics Concerns:**

no concern

**Strength And Weaknesses:**

Strength: Up to my knowledge the presented algorithm is novel and correct. The paper is overall well-written (though, possibly the measure theoretic jargon could be explained simpler and more intuitively to increase the impact).

Weaknesses:
1. Most their experimental models are low-dimensional or toy models. It would add to the value of the paper if they could show some results on more complicated models (e.g. multimodal models) and high-dimensional models (to represent the role of the dimension on the attractive/repulsive forces.
2. The importance (and novelty) of their theoretical findings is not clearly explained.
3. The convergence of the proposed algorithm to the target distribution is not guaranteed meaning the algorithm is not asymptotically unbiased.



**Summary Of The Paper:**

This paper presents a new particle based probabilistic approximate inference method that (similar to Stein Variational Gradient Descent algorithm) is based on an optimisation task involving attractive and repulsive forces. In the present work, these forces are defined differently and via a function called mollified interaction energy (MIE).
Minimising the proposed MIE function involves balancing a repulsive force to increase the distance between the particles and  an attractive force that pushes the particles to high-density regions. It is argued that the presented work might be suitable for constrained density functions.


**Summary Of The Review:**

This paper presents a new particle based inference method that minimised a function called MIE.
The algorithm is interesting but the experimental results are not thorough (see the previous section).

---

> ### Author Response · Authors · 2022-11-14
> **Response (1/2)**
>
> Thank you for your valuable feedback. Please find below our response to each of your comments/questions.
>
> **Most [of] their experimental models are low-dimensional or toy models. It would add to the value of the paper if they could show some results on more complicated models (e.g. multimodal models) and high-dimensional models (to represent the role of the dimension on the attractive/repulsive forces.**
>
> We emphasize that two of our experiments, Bayesian logistic regression, and fairness Bayesian neural networks, are high-dimensional. For Bayesian logistic regression, the dimension goes up to 61, and for the fairness experiment, the number of parameters in the Bayesian neural network is 4452, which is the dimension we are sampling in. Note that computing Wasserstein distances in high dimensions can be unreliable by the well-known $O(N^{-1/n})$ sample complexity of Wasserstein distances where $n$ is the dimension. Hence for instance for the fairness experiment we use the trade-off curves in Figure 4 between fairness and accuracy on the test data as a way to compare the performance.
>
> To help address your concerns, we have added three additional experiments with multimodal distributions in the revised paper. The results are summarized in Figure 9, Figure 10, and Figure 11. In Figure 9, we see that MIED excels at sampling a mixture of Gaussians constrained in a box. In Figure 10, we see that even with very challenging constraints, MIED can still perform well. In Figure 11, we see the effectiveness of MIED on sampling from a von Mises-Fisher distribution that has two modes constrained to a unit sphere.
>
> **The importance (and novelty) of their theoretical findings is not clearly explained.**
>
> We have revised the draft to make our theoretical contributions more apparent.
>
> We consider the importance and novelty of our theoretical findings to be two-fold.
> * Compare to SVGD and other methods based on RKHS theory, as detailed in the second last paragraph in the introduction, we use a fundamentally different and novel analysis based on mollifiers that give rise to kernels with diminishing bandwidths. The idea of using diminishing bandwidths for SVGD-like methods has not been explored before. While SVGD-like methods require special treatment [Shi et al. 2021] for constrained sampling that is difficult to generalize to arbitrary constraints like the ones we considered, our analysis results in a unified energy MIE that is the same for constrained and unconstrained domains. Furthermore, the current work proposes a new way of sampling by finite-dimensional optimization, unlike SVGD. While KSDD also samples using finite-dimensional optimization, it is only viable for the unconstrained case and our method performs consistently better and is faster/requires only first-order derivatives.
> * Compared to a different line of work summarized in [Borodachov et al. 2019] that focuses primarily on hypersingular Riesz energy, as written in our related works section and discussed further in Appendix B, we use variational analysis to derive our novel convergence results of the MIEs which is not limited to Riesz kernels and allows non-compact domains (a limitation of ours is that we require the domain to be full-dimensional and we believe this can be addressed in future work). Our measure-theoretical techniques result in a clear proof of the convergence to $\chi^2$ divergence (Theorem 3.3) as well as a proof of $\Gamma$-convergence (Theorem 3.6) using Fourier transforms of measures, which ensures the convergence of minimizers to the target measure. Additionally, in Section 3.5 we study the gradient flow of MIE and show its subdifferential converges to that of $\chi^2$ divergence as well as displacement convexity at the target measure. These close connections between interaction energies and certain probability divergence like $\chi^2$ divergence have not been established before for energies considered in [Borodachov et al. 2019].

---

> ### Author Response · Authors · 2022-11-14
> **Response (2/2)**
>
> **The convergence of the proposed algorithm to the target distribution is not guaranteed meaning the algorithm is not asymptotically unbiased.**
>
> In Theorem 3.6, we show the minimizers of MIEs converge to the target distribution. Quantifying the bias of MIED due to a positive mollifier parameter $\epsilon$ is a hard question that we plan to investigate in future work as acknowledged in the conclusion.
>
> Biases due to discretization are universal for many sampling methods. In Section A.4.3 we added a descent lemma to investigate the effect of time discretization of MIED. As discussed in our response to reviewer FquR, LMC (Langevin Monte Carlo) has bias coming from using a fixed step size (and to obtain an unbiased sampler we need a Metropolis step, resulting in MALA algorithm but it suffers from a high rejection rate in high dimensions), while non-asymptotic finite-particle theoretical results of SVGD and KSDD are still largely missing. Quantifying the bias due to discretization for MIED is also an important future work direction.
>
>
> **Also the time-complexity analysis is missing**
>
> We added Algorithm 1 in Appendix C and commented on the time complexity of each update iteration. To summarize, the time complexity is $O(N^2)$ at each iteration, the same as SVGD. Empirically we also found MIED to take about the same time as SVGD in all experiments, while KSDD takes significantly longer since it requires higher-order derivatives.

---

> ### Author Response · Authors · 2022-12-08
> **Could you please check our response?**
>
> Dear Reviewer,
>
> Since only a few days remain in the discussion period, we would appreciate it if you can check and reply to our response soon. This will give us time to address further questions that you may have before the end of the discussion period.
>
> To highlight our revision in regard to your comments, 1) we have added three additional experiments with multimodal distributions in the revised paper (Figure 9-11); 2) we revised the theoretical sections to make our contribution clearer; 3) in Section A.4.3 we proved a descent lemma to investigate the effect of time discretization; 4) in Appendix C we included an algorithm box for MIED and compared it with that of SVGD.
>
> If our response adequately addresses your concerns, please consider raising the score of our submission. Thank you for your time.

---

### Official Review · Reviewer_cXXg · 2022-10-25

**Confidence:** 3
**Correctness:** 3
**Technical Novelty And Significance:** 3
**Empirical Novelty And Significance:** 3
**Recommendation:** 8

**Clarity, Quality, Novelty And Reproducibility:**

The contribution is novel and solid with some potential applications. However, the main contributions are presented in a clear and understandable way.

**Details Of Ethics Concerns:**

No ethics concerns.

**Strength And Weaknesses:**

Strengths:

- The motivation of the paper is clear and interesting: sampling from a constrained domain either contain expensive numerical subroutines or requires explicit formulas for quantities, which are invalid in most of the cases.
-The paper is overall well-written and the authors tries to make the main idea easy to follow.

Weakness:
- Since the performance in uncontrained sampling is close to the previous methods, the authors could put more emphasize on the contrained sampling part to show the effectiveness of the method. Some of the constrained sampling results could be moved to the appendix.
- How to choose the hyperparameter s in practice? Is the method sensitive to s?

Typo:  $\int_x f(x)^p >0$.


**Summary Of The Paper:**

The paper introduces algorithms for sampling constrained distributions. The authors introduce an algorithm MIED that transform this sampling problem into an optimization problem, and practically, to optimize the mollified interaction energy in a first-order particle-based solution. They also proved that by minimizing the energy function, the MIE coverges to the target measure in the sense of chi-square divergence. The authors experimentally validate their algorithms on simulated data to do unconstrained sampling and constrained samplign.

**Summary Of The Review:**

This paper addresses an interesting problem and its development looks very exciting. The work is very complete and addresses both theoretical and experimental aspects of the proposed methods. The theory proposed looks sound with a strong background and the experiments look convincing.

---

> ### Author Response · Authors · 2022-11-14
> **Response**
>
> Thank you for your positive review. We appreciate it that you like both the theoretical and the experimental aspects of our work. Please find below our response to each of your comments/questions.
>
> **Since the performance [for] uncon[s]trained sampling is close to the previous methods, the authors could put more [emphasis] on the [constrained] sampling part to show the effectiveness of the method. Some of the constrained sampling results could be moved to the appendix.**
>
> We agree the biggest highlight of our method compared to past methods is its flexibility with constraints handling. To further illustrate this point, we have conducted three more constrained sampling experiments:
> In Section D.4 we study different choices of $s$ for MIED with Riesz mollifiers on a constrained sampling problem with four mixtures of Gaussians.
> In Section D.5, Figure 10, we run MIED to sample uniformly on a challenging constrained set with localized initialization.
> In Section D.5, Figure 11, we sample a von Mises-Fisher distribution using MIED.
> We have added texts in the main paper to refer to these experiments.
>
> Regarding unconstrained sampling, MIED consistently results in lower Wasserstein distances compared to SVGD and KSDD. This is coherent with the observation that MIED generates more evenly-spaced samples and it implies that MIED will result in lower Monte Carlo errors compared to SVGD/KSDD.
>
> On the theoretical side, proving results for unconstrained domains is *far more challenging* than proving results for compact constrained domains. For instance, the foremost technical obstacle in proving various notions of convergence is the fact that the target density may not be bounded from below (e.g. any Gaussian): since $p^{-1/2}$ appears in $W_\epsilon$, the integrand of MIE, as a result, MIE can blow up easily. Yet this is automatic if we are working with a compact domain since continuous non-vanishing functions are bounded below on compact sets. Our Theorem 3.3 and Theorem 3.6 are both valid for unconstrained domains. For both theorems, special measure theoretical techniques are necessary to get around the non-compactness of the domain: for Theorem 3.3 we need the theory of mollifiers and $L^p$ spaces, whereas for Theorem 3.6 we need to work in the Fourier domain and use Bochner’s Theorem. For comparison, the results from [Borodachov et al. 2019] on hypersingular Riesz energy (discussed in Section B) all assume the domain is compact.
>
>
> **How to choose the hyperparameter $s$ in practice? Is the method sensitive to s?**
>
> We study different choices of $s$ for MIED in the new Section D.4. Interestingly, the best-performing $s$ in this experiment is slightly bigger than the dimension $n$. Although we did all the experiments in the main text using $s = n + 10^{-4}$, we think there is room for hyperparameter searching over $s$ to further improve the performance of MIED.

---

### Official Review · Reviewer_FquR · 2022-10-27

**Confidence:** 3
**Correctness:** 3
**Technical Novelty And Significance:** 3
**Empirical Novelty And Significance:** 3
**Recommendation:** 5

**Clarity, Quality, Novelty And Reproducibility:**

The writing in general is good. The clarity on the difference between MIED with previous methods such as SVGD could be elaborated more on the algorithm part.


**Strength And Weaknesses:**

This paper is well-written and clearly structured. The proposed method is more general than the previous ones compared in the paper. The experimental results also show the advantage of MIED in constrained and unconstrained sampling problems. The main weakness of the paper is its lack of discussion about the comparison of MIED with LMC-based algorithms both theoretically and empirically. The theoretical results in this paper are mainly asymptotic and it is not clear whether MIED will have good convergence properties in various empirical applications.


**Summary Of The Paper:**

This paper studies the problem of sampling from a target density without knowing its normalization constant. It focuses on one class of variational inference methods called Stein variational gradient descent (SVGD), which formulates the sampling problem into an optimization problem that minimizes the KL distance between the target distribution and a distribution that defines the mollified interaction energy. Based on it, the authors propose the mollified interaction energy descent (MIED) method to run gradient descent on the minimization problem. They proved that the exact version of MIED asymptotically converges to the target distribution. Empirical evaluations are provided to compare MIED with previous methods.


**Summary Of The Review:**

As described in the previous sections, I think this paper is studying a very interesting problem and the algorithm proposed here seems to be competitive over existing methods, especially SVGD. However, it is not discussed in full detail what the difference between them at an algorithmic level is. It would be better to have some discussion on the difference between the algorithm design in SVGD and MIED in the algorithm section (Section 4).

Moreover, the theoretical results in this paper are all in the asymptotic sense. Moreover, the problem in (2) is not directly solvable and the paper proposes to minimize a discrete version of (2) in practice instead, which brings additional approximation error. It is not clear how large these errors could affect the sampling error in the final results. It is a natural question that does the proposed algorithm enjoy any non-asymptotic convergence? This differs from the LMC based approaches, where finite-time and fast convergence could be established for sampling from log-concave and non-log-concave densities (see [1, 2, 3] for example). Then what are the difference and advantage of using MIED instead of other LMC methods?

[1] Dalalyan, Arnak S. "Theoretical guarantees for approximate sampling from smooth and log concave densities." Journal of the Royal Statistical Society: Series B. 2017.
[2] Zou et al. "Faster convergence of stochastic gradient Langevin dynamics for non-log-concave sampling." Uncertainty in Artificial Intelligence. 2021.
[3] Muzellec et al. "Dimension-free convergence rates for gradient Langevin dynamics in RKHS." Conference on Learning Theory. 2022.

In Figure 3, why do the samples from SVGD collapse?

Although the experiments show that MIED can deal with different constraints unlike SVGD, could you explain what causes the performance drop of MIED in Figure 6?

What is the dataset used in the training of fair Bayesian neural networks?

---

> ### Author Response · Authors · 2022-11-14
> **Response (1/2)**
>
> Thank you for your valuable feedback. Please find below our response to each of your comments/questions.
>
> **The main weakness of the paper is its lack of discussion about the comparison of MIED with LMC-based algorithms both theoretically and empirically. What are the difference and advantage[s] of using MIED instead of other LMC methods?**
>
> We have revised the related works section to include more discussion of LMC methods. We have also compared against mirror LMC [Zhang et al. 2020] in Figure 3 for the uniform sampling experiment.
>
> The question of LMC vs. methods based on interacting particles (SVGD/KSDD/MIED) is a classical and difficult one. Our answers here are by no means exhaustive. LMC methods, as well as SVGD/KSDD/MIED all, come from certain time and spatial discretizations of the gradient flow of specific energies: KL divergence for LMC/SVGD, KSD (kernel Stein discrepancy) for KSDD, and MIE for MIED. The difference between LMC and particle-based sampling algorithms (SVGD/KSDD/MIED) is that LMC is stochastic and requires a single particle (or multiple non-interacting particles) while the others are deterministic with interacting particles.
>
> While LMC is typically faster to simulate and easier to parallelize as there is no interaction, it is a well-known fact that LMC introduces bias (i.e. it converges to an approximate target distribution) if a fixed step size is used. For SVGD, however, even with a fixed step size, if there are infinitely many particles, it remains unbiased (see the descent lemmas in [1][2] and the metrization of weak convergence by KSD as shown in [3]). Moreover, it has been recently shown that these interacting particle-based algorithms can lead to particle configurations that attain a lower Monte Carlo error than MCMC algorithms [4].
>
> Regarding constraint sampling, LMC (similar to MSVGD/SVMD) needs to be equipped with a mirror map which significantly limits the type of constraints it can handle [Zhang et al. 2020]. For instance, it cannot be applied to the fairness Bayesian neural network problem in Figure 4 (where the constraint requires evaluating a neural network) nor can it handle non-convex constraints in Figure 10 and Figure 11.
>
> For scenarios where a mirror map exists such as a box constraint, LMC does not produce evenly spaced samples as MIED: we added the result of mirror LMC in Figure 3 and as shown in the left plot of the revised Figure 3, running LMC leads to a bigger Wasserstein-2 distance than MIED.
>
> An additional advantage of MIED over LMC or SVGD is that it is optimization-based (i.e. it is an optimization over an objective of discrete particles). Hence monitoring convergence is easy and faster optimizers with adaptive step sizes like Adam can be used (Adam/RMSProp is still commonly used for SVGD but there is no theoretical justification for such a choice). Compared to KSD (which is also optimization-based), MIED is first-order and empirically it performs better in our experiments.
>
>
> [1] Liu, Qiang. "Stein variational gradient descent as gradient flow." Advances in neural information processing systems 30 (2017).
>
> [2] Korba, Anna, et al. "A non-asymptotic analysis for Stein variational gradient descent." Advances in Neural Information Processing Systems 33 (2020): 4672-4682.
>
> [3] Gorham, Jackson, and Lester Mackey. "Measuring sample quality with kernels." International Conference on Machine Learning. PMLR, 2017.
>
> [4] Xu, Lantian, Anna Korba, and Dejan Slepcev. "Accurate Quantization of Measures via Interacting Particle-based Optimization." International Conference on Machine Learning. PMLR, 2022.

---

> ### Author Response · Authors · 2022-11-14
> **Response (2/2)**
>
> **The theoretical results in this paper are mainly asymptotic and it is not clear whether MIED will have good convergence properties in various empirical applications. Does the proposed algorithm enjoy any non-asymptotic convergence? This differs from the LMC-based approaches…**
>
> To our knowledge, non-asymptotic finite-particle theoretical results of SVGD and KSDD are still largely missing. In the revised Section A.4.3, we have proved a descent lemma for MIE with time discretization (infinite particles). Compared to a similar result in [Korba et al. 2021], we require fewer regularity assumptions on the target density. Our descent lemma is only a local result (similar to [Korba et al. 2021]) as MIE is only displacement convex at the target density as $\epsilon \to 0$. Proving a finite-particle descent lemma and finding non-asymptotic bounds for the discrete minimizers of MIE is an important future direction.
>
>
> **It would be better to have some discussion on the difference between the algorithm design in SVGD and MIED in the algorithm section (Section 4)**
>
> In the revised Section C, we have derived the particle-update formula of MIED (Algorithm 1) and compared it with that of SVGD (Section C.1). Notably, our formulation results in scaling factors that stabilize the optimization and adaptively put more weight on nearby pairs of particles and the particles in low-density regions.
>
> **In Figure 3, why do the samples from SVGD collapse?**
>
> We investigate the collapsing issue of SVGD with an adaptive kernel in the revised Section D.2. To summarize, this happens when the kernel width becomes too big and the same pathological phenomenon occurs for unconstrained sampling as well (see Figure 7). Our theory of MIED recommends choosing small kernel widths in practice (Theorem 3.3), while the theory of SVGD has no such direct implication.
>
> We emphasize that SVGD/KSDD is not designed for constrained sampling: Stein’s identity would fail if the target density is non-vanishing on the boundary of the domain (see, e.g., the discussion under (1) in [Liu et al. 2016]).
>
> **Although the experiments show that MIED can deal with different constraints unlike SVGD, could you explain what causes the performance drop of MIED in Figure 6?**
>
> We agree that as shown in Figure 8 (previously Figure 6), MSVGD and SVGD achieve better numbers than MIED for sampling a 20-dimensional Dirichlet distribution ($W_2$ distances are only slightly better than MIED). The performance of MIED does not “drop”---it still results in well-distributed samples—but rather MSVGD and SVMD perform well because the entropic mirror map used here is a well-tested choice for simplex constraint. We suspect MIED achieves noticeably higher energy distance due to the bias introduced by using finite particles.
>
> We comment that MSVGD and SVMD are inflexible with constraint handling (see our response to your first point): they can perform poorly even with a simple box constraint (Figure 3).
>
>
> **What is the dataset used in the training of fair Bayesian neural networks?**
>
> Thank you for pointing it out. We use the Adult Income dataset [Kohavi et al. 1996]. This has been clarified in the revised draft in Section 5.

---

> ### Author Response · Authors · 2022-12-08
> **Could you please check our response?**
>
> Dear Reviewer,
>
> Since only a few days remain in the discussion period, we would appreciate it if you can check and reply to our response soon. This will give us time to address further questions that you may have before the end of the discussion period.
>
> To highlight our revision in regard to your comments, 1) we added more discussion and experiments for LMC; 2) we added a new section C to detail the algorithmic differences between SVGD and MIED; 3) we investigated the collapsing issue of SVGD in Section D.2.
>
> If our response adequately addresses your concerns, please consider raising the score of our submission. Thank you for your time.

---

### Author Response · Authors · 2022-11-14
**Updates on the revision**

We thank all the reviewers for their time writing the helpful initial reviews. We have revised our submission accordingly in the following main aspects. *The reference numbers used below are all based on the newest revision.*

- We have revised the writing of the paper and fixed typos. We have thoroughly improved the clarity of mathematical definitions, theoretical statements, and proofs.
- We have proved a much stronger version of Gamma-convergence (previously Proposition 3.4(c), now standalone Theorem 3.6). We remove all previous assumptions on the sequence of minimizers $\{\mu_\epsilon\}$ (e.g. equicontinuous densities) and only require the kernel is i.s.p.d. The proof is based on a new interpretation of MIE in the Fourier domain (Proposition A.16) and an inverse version of the Plancherel theorem. We believe the proof (provided in Section A.3) is theoretically interesting on its own as it demonstrates that working in the Fourier domain can circumvent the integrability issues caused by taking $\epsilon \to 0$ when working with mollifiers.
- We have proved a descent lemma with time discretization in Appendix A.4.3. Compared to a similar lemma by [Korba et al. 2021], we require fewer regularity assumptions.
- We have added algorithmic details in Appendix C in the revised appendix, compared them with the updates from SVGD, and added details for the dynamic barrier method generalized to handling multiple constraints.
- Several additional sections are added in Appendix D:
    - In Section D.2 we investigate the issue of collapsed samples when the kernel width is too big.
    - In Section D.4 we study the effect of $s$ for MIED with Riesz mollifiers.
    - In Section D.5 we test MIED on challenging constraints and multi-model distributions (von Mises-Fisher).

---

### Decision · Program_Chairs · 2023-01-20

**Decision:**

Accept: poster

**Justification For Why Not Higher Score:**

Although this paper proposes a new sampling method in the line of the stein discrepancy type methods, its theoretical guarantee is yet weak. Moreover, comparison with LMC methods is also weak. For these reasons, I do not recommend this paper to spotlight presentation.

**Justification For Why Not Lower Score:**

The proposed algorithm is interesting and sufficiently novel. The proposal will be beneficial to the community. The writing is also clear. Then, I recommend acceptance to the conference.

**Metareview: Summary, Strengths And Weaknesses:**

This paper proposes a new method called mollified interaction energy descent (MIED) that can be used for sampling particles from densities for which it is difficult to obtain the normalizing constant. MIED minimizes an energy functional that has both attraction and repulsion terms. It is theoretically shown that the proposed energy function can be seen as a mollified version of the chi-square distance and by letting the tuning parameter $\epsilon$ converge to zero, the energy functional converges to the chi-square distance. It is also useful to sample from a constraint domain. The proposed algorithm is examined in some numerical experiments so that its effectiveness is justified.

The proposed algorithm is indeed novel. The writing is clear. The proposed method is well motivated with a theoretical guarantee.
On the other hand, the paper also has some drawbacks. First, there is no theoretical guarantee about finite particle approximation errors while existing MCMC type methods have strong theoretical back-ground on this issue. The algorithmic convergence is also not well justified. It is possible that the particles could stack at some region which could be far away from the optimal configuration due to gradient vanishing. Second, the numerical experiments are mainly conducted to compare with SVGD/KSDD methods, but comparison with LMC type methods are not thorough. While the authors added comparisons with LMC and some related discussions, it is better if there is more convincing discussions.

The authors could put more emphasize on the fact that the algorithm is effective also for constraint sampling.

Although there are some weakness as stated above, this paper gives an interesting algorithm which is sufficiently novel. Its theoretical guarantee could be explored in depth in the future.

**Note From Pc:**

if the above contains the word "oral" or "spotlight" please see: "oral" presentation means -> notable-top-5% and "spotlight" means -> notable-top-25%. As stated in our emails, we are disassociating presentation type from AC recommendations